# Hierarchical Information Flow for Generalized Efficient Image Restoration

## Abstract

While vision transformers show promise in numerous image restoration (IR) tasks, the challenge remains in efficiently generalizing and scaling up a model for multiple IR tasks. To strike a balance between efficiency and model capacity for a generalized transformer-based IR method, we propose a **h**ierarchical **i**nformation flow mechanism for **i**mage **r**estoration, dubbed **Hi-IR**, which progressively propagates information among pixels in a bottom-up manner. Hi-IR constructs a hierarchical information tree representing the degraded image across three levels. Each level encapsulates different types of information, with higher levels encompassing broader objects and concepts and lower levels focusing on local details. Moreover, the hierarchical tree architecture removes long-range self-attention, improves the computational efficiency and memory utilization, thus preparing it for effective model scaling. Based on that, we explore model scaling to improve our method's capabilities, which is expected to positively impact IR in large-scale training settings. Extensive experimental results show that Hi-IR achieves state-of-the-art performance in seven common image restoration tasks, affirming its effectiveness and generalizability.

## 1 Introduction

Image restoration (IR) aims to improve image quality by recovering high-quality visuals from observations degraded by noise, blur, and downsampling. To address this series of inherently ill-posed problems, numerous methods have been developed primarily for a single degradation, including convolutional neural networks (CNNs) (Dong et al., 2014; Kim et al., 2016; Lim et al., 2017), vision transformers (ViTs) (Chen et al., 2021; Liang et al., 2021; Li et al., 2023a), and state space models (Mamba) (Gu & Dao, 2023; Guo et al., 2024). However, the intricate and varied nature of degradation presents formidable challenges to the prevailing IR methodologies. In particular, several coupled problems remain for general IR:

- First, *there is a lack of a generalized computational mechanism for efficient IR.* A general IR framework needs to deal with images with varying characteristics, such as different types and intensities of degradation, as well as varying resolutions. Techniques designed for specific IR tasks might not apply to other problems. Simply combining computational mechanisms designed for different IR tasks does not necessarily result in an efficient solution. Thus, it is a challenge to design a mechanism that is both efficient and capable of generalizing well to different IR tasks.

- Second, *there is no systematic approach for guiding model scaling.* Current image restoration networks are typically limited to 10-20M parameters. Addressing multiple degradations often requires increasing the model capacity by scaling up the model size. Yet, diminished model performance is observed by simply scaling up the model. Therefore, the challenge of systematically scaling up IR models remains unresolved.

- Third, *it is still unclear how well a single model can generalize across different IR tasks.* Existing approaches tend to focus on either a single task or a subset of IR tasks. The generalizability of a single model across a broader range of IR tasks has to be thoroughly validated.

This paper addresses the aforementioned questions in Sec. 3, Sec. 4. and Sec. 5, respectively. We propose a hierarchical information flow principle designed specifically for general IR tasks. This principle establishes relationships between pixels on multiple levels and progressively aggregates

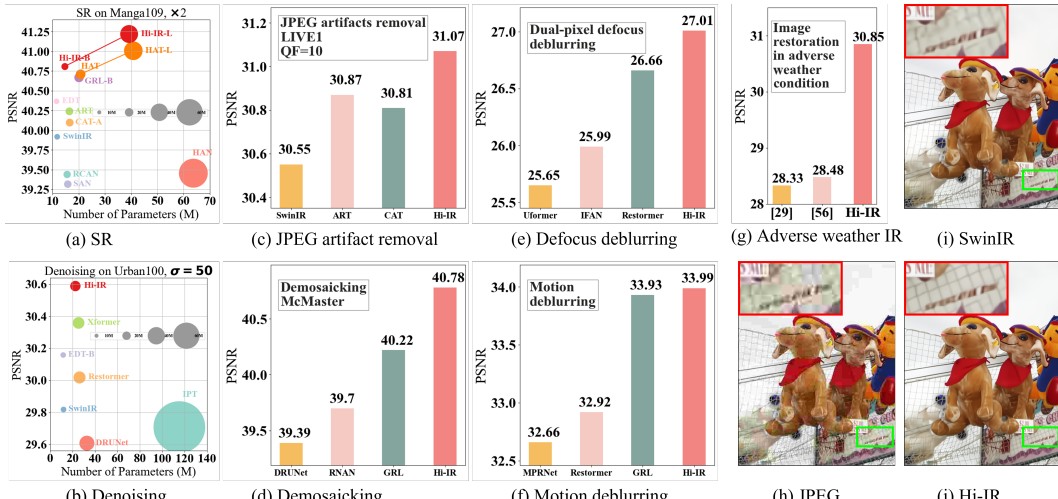

Figure 1: The proposed Hi-IR is notable for its efficiency and effectiveness (a)-(b), generalizability across seven image restoration tasks (a)-(g), and improvements in the visual quality of restored images (h)-(j).

information across multiple levels, which is essential for general IR. Compared with existing approaches such as convolution (Zhang et al., 2018c), global attention (Chen et al., 2021), and window attention (Li et al., 2023a), hierarchical information flow balances complexity with the efficiency of comprehending global contexts, ensuring an optimized process for integrating information across various scales and regions. The underlying design principle opens the door to different realizations. Considering the effectiveness and efficiency for image modeling, we propose a new architecture based on a three-level **h**ierarchical **i**nformation flow mechanism for **i**mage **r**estoration (*i.e.,* **Hi-IR**). Hi-IR employs a series of progressive computational stages for efficient information flow. The first-level (L1) computational block works within individual patches, fostering local information exchange and generating intermediate node patches. Then, a second-level (L2) block works across the intermediate node patches and allows for the effective propagation of information beyond the local scope. As a final step, the third-level (L3) information flow block bridges the gaps between the isolated node patches from the first two stages.

Motivated by the scaling law (Brown et al., 2020; Touvron et al., 2023; Kang et al., 2023; Saharia et al., 2022; Yu et al., 2024), we scale up the model to enhance the model capacity. We analyze the reason why it is difficult to scale up IR models. As a remedy to the notorious problem (Lim et al., 2017; Chen et al., 2023), this paper proposes three strategies that systematically encompass model training, weight initialization, and model design to enable effective model scaling.

This paper validates the generalizability of the proposed hierarchical information flow mechanism through rigorous experiments on multiple aspects. First, we investigate the performance of the model trained for a specific degradation type and intensity, including downsampling, motion blur, defocus blur, noise, and JPEG compression. Second, we validate that the model can handle a single degradation type with multiple intensities. Furthermore, we demonstrate that a single model can generalize effectively across multiple tasks, validating its versatility. Our main contributions are summarized as follows:

- We introduce a novel hierarchical information flow principle for image restoration, which facilitates progressive global information exchange and mitigates the curse of dimensionality.

- We propose Hi-IR, a compact image restoration model guided by the design principle, to propagate information for image restoration efficiently.

- We examine the challenge of training convergence for model scaling-up in IR and propose mitigation strategies.

- Extensive experiments demonstrate the generalizability of the proposed hierarchical information flow mechanism. The proposed Hi-IR consistently outperforms state-of-the-art image restoration methods for multiple tasks.

## 2 RELATED WORK

**Image Restoration** focuses on recovering high-quality images from their degraded counterparts. As a challenging problem, IR has captured substantial interest in academic and industrial circles, leading to practical applications such as denoising, deblurring, super-resolution (SR), and so on. The landscape of IR has shifted with the evolution of deep learning and the increased availability of computational resources, notably GPUs. Neural network-based pipelines, fueled by advancements in deep learning, have supplanted earlier model-based solutions (Richardson, 1972; Liang et al., 2021; Li et al., 2023b). Numerous CNN models have been proposed (Anwar & Barnes, 2020; Li et al., 2022b; Dong et al., 2014; Zhang et al., 2017a) for different IR tasks. However, despite their effectiveness, CNNs have been found to struggle in propagating long-range information within degraded input images. This challenge is attributed to the limited receptive field of CNNs, which, in turn, constrains the overall performance of CNN-based methods (Chen et al., 2022b; Zhang et al., 2022; Li et al., 2023a).

**Vision Transformer-based Models for IR** have been proposed to address the problem of global information propagation inspired by the success of Transformer architecture in machine translation (Vaswani et al., 2017) and high-level vision tasks (Dosovitskiy et al., 2020). Specifically, IPT (Chen et al., 2021) applies ViTs for IR. Despite promising results, it is difficult to use full-range self-attention within the ViTs because the computational complexity increases quadratically with the image size. As a remedy, numerous methods explore ViTs in an efficient yet effective manner. In particular, SwinIR (Liang et al., 2021) conducts multi-head self-attention (MSA) window-wise. A shift operation is applied to achieve the global interactive operation (Liu et al., 2021). Uformer (Wang et al., 2022) proposes to propagate much more global information with a UNet structure but still with window self-attention. Other methods (Zamir et al., 2022; Chen et al., 2022b; Ren et al., 2024) re-design the attention operation with much more exquisite efforts, such as cross-covariance across channel dimensions (Zamir et al., 2022), rectangle-window self-attention (Li et al., 2021), sparse self-attention Huang et al. (2021), and graph-attention (Ren et al., 2024), spatial shuffle (Huang et al., 2021), and random spatial shuffle Xiao et al. (2023). However, these transformer-based solutions cannot balance the ability to generalize to multiple IR tasks and the computational complexity of global modeling. In this paper, we propose a general and efficient IR solution which hierarchically propagates information in a tree-structured manner, simultaneously incorporating inputs from lower and higher semantic levels.

## 3 METHODOLOGY

### 3.1 MOTIVATION

This paper aims to propose a general and efficient IR framework. Before presenting technical details, we discuss the motivation behind the proposed hierarchical information flow mechanism.

In this work, we demonstrate the pivotal role of the information flow in decoding low-level features, which become more pronounced with the introduction of ViTs. CNNs employ successive convolutions that inherently facilitate progressive information flow beyond local fields. In contrast, image restoration transformers typically achieve information flow via self-attention across manually partitioned windows, combined with a window-shifting mechanism. When the flow of contextual information between different regions or features within an image is restricted, a model's ability to reconstruct high-quality images from low-quality counterparts is significantly hindered. This effect can be observed by deliberately isolating the information flow in Swin transformer. In Tab. 1, the flow of information across windows is prohibited by removing the window-shifting mechanism, which leads to a decrease in PSNR on the validation datasets (specifically, a 0.27 dB drop for DF2K training, and a 0.23 dB drop for LSDIR training). The obvious reductions indicate that information isolation degrades the performance of IR techniques, likely because the algorithms are deprived of the contextual clues necessary for accurately reconstructing finer image details.

Secondly, we observe that information propagation on fully connected graphs is not always necessary or beneficial for improving the performance of the IR networks (Chen et al., 2021; Zamir et al., 2022). As ViTs generate distinct graphs for each token, early attempts to facilitate global information dissemination led to the curse of dimensionality, causing quadratic growth in computational complexity with token increase (Wang et al., 2020; Liu et al., 2021). Subsequent attention

Table 1: Removing shifted windows leads to degraded SR performance. PSNR is reported on Urban100 dataset for $4\times$ SR.

| Training Dataset | Window Shift | |
|---|---|---|
| | Yes | No |
| DF2K (Agustsson & Timofte, 2017) | 27.45 | 27.18 (-0.27) |
| LSDIR (Li et al., 2023b) | 27.87 | 27.64 (-0.23) |

Table 2: Plateau effect of enlarged window size reported on Urban100 for $4\times$ SR. Window size larger than 32 is not investigated due to the OOM issue.

| Window size | PSNR | PSNR gain | GPU Mem. | Computation |
|---|---|---|---|---|
| 8 | 27.42 | 0.00 | 14.63GB | $\times 1$ |
| 16 | 27.80 | +0.38 | 17.22GB | $\times 4$ |
| 32 | 28.03 | +0.22 | 27.80GB | $\times 16$ |

(a) CNN Based

(b) Global Attention Based

(c) Window-Attention Based

(d) Tree Structure Attention (Ours)

Figure 2: Illustration of information flow principles. The colors represent local information, with their blending indicating propagation beyond the local region. (a) The CNN-based. (b) The original ViTs based. (c) Window attention based. (d) The proposed hierarchical information flow prototype.

mechanisms, building graphs based on windows, achieve better IR results. However, the benefits of expanding the window size tend to plateau. Tab. 2 shows the effect of window size versus performance. The quality of the reconstructed images improves as the window size grows from 8 to 32, evident from rising PSNR values. Yet, with larger windows, the gains decrease, accompanied by a sharp increase in memory footprint and computational demands, resulting in a plateau effect. This prompt a reassessment of the information propagation mechanism on large windows. The challenge lies in balancing the scope and the complexity of window attention while enhancing global information propagation efficiency.

**Effective information flow.** The above analysis emphasizes the crucial role of effective information flow in modern architectural designs. CNN-based methods propagate information slowly within a small region covered by the filter (Fig. 2(a)). A large receptive field has to be achieved by the stack of deep layers. Global attention based ViT propagates information directly across the whole sequence with a single step. However, the computational complexity grows quadratically with the increase of tokens (Fig. 2(b)). To address this problem, window attention in Fig. 2(c) propagates information across two levels but still has a limited receptive field even with shift operation.

To facilitate fast and efficient information flow across the image, we propose a hierarchical information flow principle shown in Fig. 2(d). In this model, information flows progressively from the local scope, aggregated in several intermediate levels, and disseminated across the whole sequence. This new design principle is more efficient in that it enables a global understanding of the input sequence with several operations. Moreover, the actual implementation of the tree structure such as the depth of the tree can be configured to ensure computational efficiency. One realization in this work is a three-level information flow model.

## 3.2 HIERARCHICAL TREE-STRUCTURED INFORMATION FLOW

As shown in Fig. 3(a) - (c), the hierarchical tree-structured information flow mechanism consists of three levels and aims to effectively model both the local and the global information for a given feature $X \in \mathbb{R}^{H \times W \times C}$ efficiently. We denote the information within $X$ as $l_0$ level meta-information.

**L1 information flow attention** is achieved by applying MSA to the input feature $X$ within a $p \times p$ patch. To facilitate the MSA, the input feature is first partitioned into local patches, leading to $X' \in \mathbb{R}^{\frac{HW}{p^2} \times p^2 \times C}$. Then feature $X'$ is linearly projected into query $(Q^{l_1})$, key $(K^{l_1})$, and value $(V^{l_1})$. Self-attention within the local patches is denoted as $Y_i^{l_1} = \text{SoftMax}(Q_i^{l_1}(K_i^{l_1})^\top / \sqrt{d})V_i^{l_1}$, where $i$ index the windows, and $d$ represents the head dimension. This process is shown in Fig. 3(a). Each node within the $Y^{l_1}$ grid represents all the $l_0$ level meta-information derived from its corresponding original window, marked by the same color.

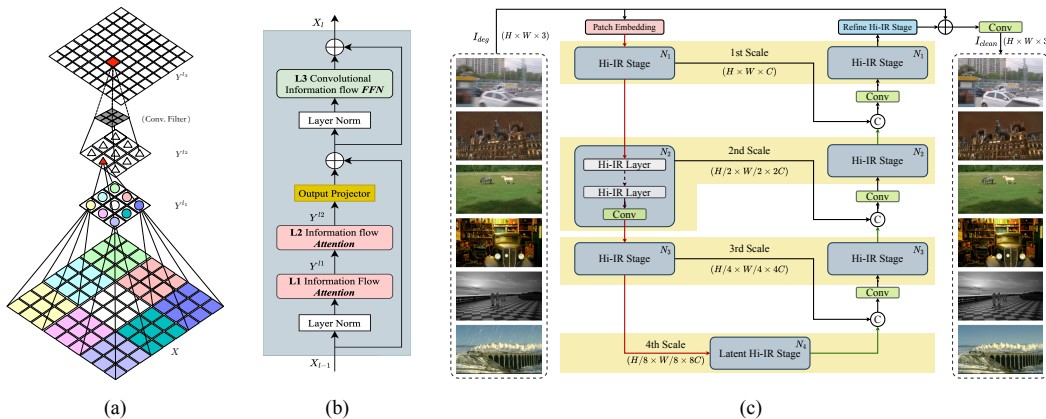

(a)  (b)  (c)

Figure 3: Illustrations of: (a) The hierarchical information flow. (b) The proposed hierarchical information flow transformer layer. (c) The overall framework of the proposed Hi-IR.

**L2 information flow attention** is achieved upon the previous $l_1$ level information $Y^{l_1}$. Despite the expanded scope of information within each grid of $Y^{l_1}$, comprehensive cross-window information propagation remains a challenge. As indicated conceptually in Fig. 2(d), 2D $s \times s$ non-overlapping local patches $p \times p$ in L1 information flow should be grouped together to form a broader $P \times P$ region for L2 information flow. Different from the previous operations (Xiao et al., 2023; Huang et al., 2021), we do not expand to the whole image in this phase due to two considerations: 1) **The computational complexity of attention in the global image can be quite high**; 2) **Not all global image information is relevant to the reconstruction of a specific pixel**. To facilitate MSA, the dispersed pixels need to be grouped together via a permutation operation. The seemingly complex operation is simplified by first reshaping the input tensor to $\hat{Y}^{l_1} \in \mathbb{R}^{\frac{H}{P} \times s \times p \times \frac{W}{P} \times s \times p \times C}$, followed by a permutation to form $(Y')^{l_1} \in \mathbb{R}^{(\frac{H}{P} \times \frac{W}{P} \times p^2) \times s^2 \times C}$. The simple permutation operation facilitates the distribution of $l_1$ information nodes across a higher level region, ensuring each window contains a comprehensive, cross-window patch-wise $l_2$ information set without hurting the overall information flow.

To better integrate the permuted information $(Y')^{l_1}$, we further project $(Y')^{l_1}$ to $Q^{l_1}$, $K^{l_1}$, and $V^{l_1}$. And the second MSA ($L_2$ Information flow attention in Fig. 3(b)) among patches is applied via $Y_i^{l_2} = \text{SoftMax}(Q_i^{l_1}(K_i^{l_1})^\top / \sqrt{d}) V_i^{l_1}$. As a result, the larger patch-wise global information (colorful nodes in $Y^{l_1}$) now is well propagated to each triangle node (Fig. 3) in $Y^{l_2}$.

**L3 convolutional information flow FFN** is implemented via a $3 \times 3$ convolution operation between two $1 \times 1$ convolution operations, forming the convolutional feed-forward network in this paper and outputs the third level information $Y^{l_3}$. As a result, this design not only aggregates all the channel-wise information more efficiently but also enriches the inductive modeling ability (Chu et al., 2022; Xu et al., 2021) for the proposed mechanism.

### 3.3 HI-IR LAYER

The Hi-IR layer, serving as the fundamental component for both architectures, is constructed based on the innovative tree-structured information flow mechanism (TIFM) introduced above, and the detailed structure is depicted in Fig. 3(b). For each Hi-IR layer, the input feature $X_{l-1}$ first passes through a layer normalization and two consecutive information propagation attentions. After adding the shortcut, the output $X_l'$ is fed into the convolutional feed-forward networks with another shortcut connection and outputs $X_l$. We formulate this process as follows:

$$
\begin{aligned}
X'_l &= \text{TIFM}_{\text{Att}}\left(\text{LN}\left(X_{l-1}\right)\right) + x_{l-1}, \\
X_l &= \text{TIFM}_{\text{Conv}}\left(\text{LN}\left(X'_l\right)\right) + X'_l
\end{aligned}
\tag{1}
$$

where $\text{TIFM}_{\text{Att}}$ consists of both the L1 and L2 information flow attention, $\text{TIFM}_{\text{Conv}}$ denotes the L3 convolutional information flow FFN.

## 3.4 OVERALL ARCHITECTURE

To comprehensively validate the effectiveness of the proposed method, similar to prior methods (Chen et al., 2022a; Li et al., 2023a; Ren et al., 2024), we choose two commonly used basic architectures including the U-shape hierarchical architecture shown in Fig. 3(c) and the columnar architecture shown in Fig. 7 of Appx. A.1. The columnar architecture is used for image SR while the U-shape architecture is used for other IR tasks. Specifically, given degraded low-quality image $I_{low} \in \mathbb{R}^{H \times W \times 1/3}$ (1 for the grayscale image and 3 for the color image ), it was first sent to the convolutional feature extractor and outputs the shallow feature $F_{in} \in \mathbb{R}^{H \times W \times C}$ for the following Hi-IR stages/layers. $H$, $W$, and $C$ denote the height, the width, and the channels of $F_{in}$. For the U-shape architecture, $F_{in}$ undergoes representation learning within the U-shape structure. In contrast, for the columnar architecture, $F_{in}$ traverses through $N$ consecutive Hi-IR stages. Both architectures ultimately generate a restored high-quality image $I_{high}$ through their respective image reconstructions.

## 4 MODEL SCALING-UP

Table 3: Model scaling-up exploration with SR.

| Scale | Model Size | Warm up | Conv Type | PSNR | | | | |
|---|---|---|---|---|---|---|---|---|
| | | | | Set5 | Set14 | BSD100 | Urban100 | Manga109 |
| 2× | 15.69 | No | conv1 | 38.52 | 34.47 | 32.56 | 34.17 | 39.77 |
| 2× | 57.60 | No | conv1 | 38.33 | 34.17 | 32.46 | 33.60 | 39.37 |
| 2× | 57.60 | Yes | conv1 | 38.41 | 34.33 | 32.50 | 33.80 | 39.51 |
| 2× | 54.23 | Yes | linear | 38.56 | 34.59 | 32.58 | 34.32 | 39.87 |
| 2× | 55.73 | Yes | conv3 | 38.65 | 34.48 | 32.58 | 34.33 | 40.12 |
| 3× | 15.87 | No | conv1 | 35.06 | 30.91 | 29.48 | 30.02 | 34.41 |
| 3× | 57.78 | No | conv1 | 34.70 | 30.62 | 29.33 | 29.11 | 33.96 |
| 3× | 57.78 | Yes | conv1 | 34.91 | 30.77 | 29.39 | 29.53 | 34.12 |
| 3× | 54.41 | Yes | linear | 35.13 | 31.04 | 29.52 | 30.20 | 34.54 |
| 3× | 55.91 | Yes | conv3 | 35.14 | 31.03 | 29.51 | 30.22 | 34.76 |

Table 4: Investigated weight intialization and rescaling method for model scaling-up.

| Method | Description | PSNR on Set5 | |
|---|---|---|---|
| | | 2× | 3× |
| Zero Layer-Norm | Initialize the weight and bias of LayerNorm as 0 (Liu et al., 2022b). | 38.35 | 34.81 |
| Residual rescale | Rescale the residual blocks by a factor of 0.01 (Lim et al., 2017; Chen et al., 2023). | 38.31 | 34.79 |
| Weight rescale | Rescale the weight parameters in residual blocks by a factor of 0.1 (Wang et al., 2018). | 38.36 | 34.84 |
| trunc_normal_ | Truncated normal distribution | 38.33 | 34.71 |

Existing IR models are limited to a model size of 10-20M parameters. In this paper, we develop models of medium and large sizes. However, scaling up the model size from 15M to 57M leads to an unexpected performance drop, as shown in the pink rows of Tab. 3. In addition, as shown in Appx. B, the 57M model also converges slower than the 15M model during training.

Table 5: Dot production attention *vs.* cosine similarity attention for model scaling. PSNR reported for SR.

| Scale | Attn. type | Set5 | Set14 | BSD100 | Urban100 | Manga109 |
|---|---|---|---|---|---|---|
| 2× | cosine sim | 38.43 | 34.65 | 32.56 | 34.13 | 39.69 |
| 2× | dot prod | 38.56 | 34.79 | 32.63 | 34.49 | 39.89 |
| 4× | cosine sim | 33.08 | 29.15 | 27.96 | 27.90 | 31.40 |
| 4× | dot prod | 33.14 | 29.09 | 27.98 | 27.96 | 31.44 |

**Initial attempts.** Existing methods handle this problem with weight initialization and rescaling techniques. For example, Chen et al. (2023) and Lim et al. (2017) reduce the influence of residual convolutional blocks by scaling those branches with a sufficiently small factor (0.01). Wang et al. (2018) rescale the weight parameters in the residual blocks by a factor of 0.1. Liu et al. (2022b) intialize the weight and bias of LayerNorm as 0. In addition, we also tried the truncated normal distribution to initialize the weight parameters. However, as shown in Tab. 4, none of the four methods improves the convergence and performance of the scaled models, indicating that they do work for the attention modules of the IR transformers.

**Solutions.** The initial investigation indicates that the problem can be attributed to the training strategy, the initialization of the weight, and the model design. Thus, three methods are proposed to mitigate the model scaling problem. *First*, we warm up the training for 50k iterations at the beginning. As shown in Tab. 3, this mitigates the problem of degraded performance of scaled up models, but does not solve it completely. *Secondly*, we additionally replace heavyweight $3 \times 3$ convolution (conv1 in Tab. 3) with lightweight operations besides warming up the training. Two alternatives are considered including a linear layer (linear in Tab. 3) and a bottleneck block with 3 lightweight convolutions ($1 \times 1$ conv+$3 \times 3$ conv+$1 \times 1$ conv, conv3 in Tab. 3). The number of channels of the middle $3 \times 3$ conv in the bottleneck blocks is reduced by a factor of 4. Tab. 3 shows that removing the large $3 \times 3$ convolutions leads to a much better convergence point for the large models. Considering that the bottleneck block leads to better PSNR than linear layers in most cases, it is adopted in all the other experiments. *Thirdly*, we also investigate the influence of the self-attention mechanism on

the convergence of scale-up models. Specifically, two attention mechanisms are compared including dot product attention (Liu et al., 2021) and cosine similarity attention (Liu et al., 2022b). As shown in Tab. 5, dot product self-attention performs better than cosine similarity self-attention. Thus, dot product self-attention is used throughout this paper unless otherwise stated. The rationale behind why the proposed three strategies are effective for model scaling-up is detailed in Appx. B.

## 5 EXPERIMENTS

In this section, the results of the ablation study are first reported. Then we validate the effectiveness and generalizability of Hi-IR on **7** IR tasks, *i.e.,* image SR, image Dn, JPEG image compression artifact removal (CAR), single-image motion deblurring, defocus deblurring and image demosaicking, and IR in adverse weather conditions (AWC). More details about the training protocols and the training/test datasets are shown in Appx A. The best and the second-best quantitative results are reported in red and blue. Note that † denotes a single model that is trained to handle multiple degradation levels (*i.e.,* noise levels, and quality factors) for validating the generalizability of Hi-IR.

### 5.1 ABLATION STUDIES

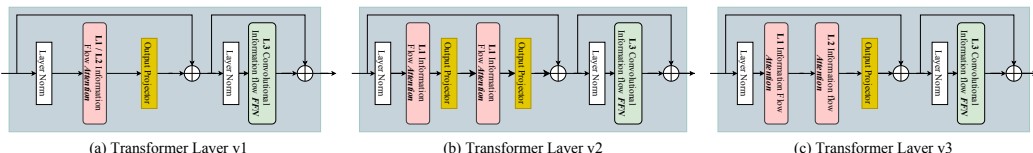

(a) Transformer Layer v1      (b) Transformer Layer v2      (c) Transformer Layer v3

Figure 4: Comparison of three types of transformer layers designed in this paper.

Table 6: Ablation study on model design with SR (reported on Set5).

| Scale | L1/L2 Version | L3 Version | | | |
|---|---|---|---|---|---|
| | | Model size [M] | | PSNR | |
| | | with L3 | w/o L3 | with L3 | w/o L3 |
| 2× | v1 | 14.35 | 11.87 | 38.34 | 38.31 |
| 2× | v2 | 19.22 | 16.74 | 38.30 | 38.22 |
| 2× | v3 | 15.69 | 13.21 | 38.37 | 38.35 |
| 2× | v4 | 17.19 | - | 38.41 | - |
| 4× | v1 | 14.50 | 12.02 | 32.89 | 32.85 |
| 4× | v2 | 19.37 | 16.89 | 32.88 | 32.77 |
| 4× | v3 | 15.84 | 13.36 | 32.92 | 32.87 |
| 4× | v4 | 17.35 | - | 32.95 | - |

Table 7: Model efficiency *vs.* accuracy for SR and Dn. PSNR is reported on Urban100 dataset.

| Task | Network | Arch. | Params [M] | FLOPs [G] | Runtime [ms] | PSNR [dB] |
|---|---|---|---|---|---|---|
| 4× SR | SwinIR (Liang et al., 2021) | Columnar | 11.90 | 215.32 | 152.24 | 27.45 |
| | CAT (Chen et al., 2022b) | Columnar | 16.60 | 387.86 | 357.97 | 27.89 |
| | HAT (Chen et al., 2023) | Columnar | 20.77 | 416.90 | 368.61 | 28.37 |
| | Hi-IR (Ours) | Columnar | 14.83 | 287.20 | 331.92 | 28.44 |
| Dn 50 | SwinIR (Liang et al., 2021) | Columnar | 11.50 | 804.66 | 1772.84 | 27.98 |
| | Restormer (Zamir et al., 2022) | U-shape | 26.13 | 154.88 | 210.44 | 28.29 |
| | GRL (Li et al., 2023a) | Columnar | 19.81 | 1361.77 | 3944.17 | 28.59 |
| | Hi-IR (Ours) | U-shape | 22.33 | 153.66 | 399.05 | 28.91 |

Extensive ablation experiments explore the following key aspects:

**Effect of L1 and L2 information flow.** One design choice for the L1/L2 information flow attentions is to decide whether to interleave them across Transformer layers or to implement them in the same layer. To validate this choice, we develop three versions, including v1 where L1 and L2 attentions alternate in consecutive layers, v2 and v3 where L1 and L2 attentions are used in the same layer (Fig. 4). Compared with v1, v2 showed reduced performance despite increased model complexity. To address this issue, we introduce v3, where the projection layer between L1 and L2 is removed and the dimension of **Q** and **K** in L1/L2 attention is reduced by half to save computational complexities. The v3 L1/L2 information flows can be conceptually unified into a single flow with an expanded receptive field. Our ablation study reveals that v3 yielded the best performance, as evidenced by the results in Tab. 6. Consequently, v3 was adopted for all subsequent experiments.

**Effect of the depth of the tree structure.** Ablation study was conducted to evaluate the effect of the tree structure's depth. In Tab. 6, the depth of the tree in the v1 model is 3. Removing the L3 information flow reduces the depth to 2, resulting in degraded image SR performance, even on the small Set5 dataset. Additionally, a v4 model was designed by adding an information flow attention beyond L2 to v3 model, creating a depth-4 tree structure. As shown in Tab. 6, this increased complexity improves SR results. Thus, well-designed deeper tree structures lead to improved model performance but with increased model complexity.

Table 8: *Classical image SR* results. Note that 10-20M models (best in light pink and second best in light cyan ) and 40M models are ranked seperately (best in red).

| Method | Scale | Params [M] | Set5 | | Set14 | | BSD100 | | Urban100 | | Manga109 | |
|---|---|---|---|---|---|---|---|---|---|---|---|---|
| | | | PSNR↑ | SSIM↑ | PSNR↑ | SSIM↑ | PSNR↑ | SSIM↑ | PSNR↑ | SSIM↑ | PSNR↑ | SSIM↑ |
| SwinIR (Liang et al., 2021) | 2× | 11.75 | 38.42 | 0.9623 | 34.46 | 0.9250 | 32.53 | 0.9041 | 33.81 | 0.9427 | 39.92 | 0.9797 |
| CAT-A (Chen et al., 2022b) | 2× | 16.46 | 38.51 | 0.9626 | 34.78 | 0.9265 | 32.59 | 0.9047 | 34.26 | 0.9440 | 40.10 | 0.9805 |
| ART (Zhang et al., 2022) | 2× | 16.40 | 38.56 | 0.9629 | 34.59 | 0.9267 | 32.58 | 0.9048 | 34.30 | 0.9452 | 40.24 | 0.9808 |
| EDT (Li et al., 2021) | 2× | 11.48 | 38.63 | 0.9632 | 34.80 | 0.9273 | 32.62 | 0.9052 | 34.27 | 0.9456 | 40.37 | 0.9811 |
| GRL-B (Li et al., 2023a) | 2× | 20.05 | 38.67 | 0.9647 | 35.08 | 0.9303 | 32.67 | 0.9087 | 35.06 | 0.9505 | 40.67 | 0.9818 |
| HAT (Chen et al., 2023) | 2× | 20.62 | 38.73 | 0.9637 | 35.13 | 0.9282 | 32.69 | 0.9060 | 34.81 | 0.9489 | 40.71 | 0.9819 |
| Hi-IR-B (Ours) | 2× | 14.68 | 38.71 | 0.9657 | 35.16 | 0.9299 | 32.73 | 0.9087 | 34.94 | 0.9484 | 40.81 | 0.9830 |
| HAT-L (Chen et al., 2023) | 2× | 40.70 | 38.91 | 0.9646 | 35.29 | 0.9293 | 32.74 | 0.9066 | 35.09 | 0.9505 | 41.01 | 0.9831 |
| Hi-IR-L (Ours) | 2× | 39.07 | 38.87 | 0.9663 | 35.27 | 0.9311 | 32.77 | 0.9092 | 35.16 | 0.9505 | 41.22 | 0.9846 |
| SwinIR (Liang et al., 2021) | 4× | 11.90 | 32.92 | 0.9044 | 29.09 | 0.7950 | 27.92 | 0.7489 | 27.45 | 0.8254 | 32.03 | 0.9260 |
| CAT-A (Chen et al., 2022b) | 4× | 16.60 | 33.08 | 0.9052 | 29.18 | 0.7960 | 27.99 | 0.7510 | 27.89 | 0.8339 | 32.39 | 0.9285 |
| ART (Zhang et al., 2022) | 4× | 16.55 | 33.04 | 0.9051 | 29.16 | 0.7958 | 27.97 | 0.7510 | 27.77 | 0.8321 | 32.31 | 0.9283 |
| EDT (Li et al., 2021) | 4× | 11.63 | 33.06 | 0.9055 | 29.23 | 0.7971 | 27.99 | 0.7510 | 27.75 | 0.8317 | 32.39 | 0.9283 |
| GRL-B (Li et al., 2023a) | 4× | 20.20 | 33.10 | 0.9094 | 29.37 | 0.8058 | 28.01 | 0.7611 | 28.53 | 0.8504 | 32.77 | 0.9325 |
| HAT (Chen et al., 2023) | 4× | 20.77 | 33.18 | 0.9073 | 29.38 | 0.8001 | 28.05 | 0.7534 | 28.37 | 0.8447 | 32.87 | 0.9319 |
| Hi-IR-B (Ours) | 4× | 14.83 | 33.14 | 0.9095 | 29.40 | 0.8029 | 28.08 | 0.7611 | 28.44 | 0.8448 | 32.90 | 0.9323 |
| HAT-L (Chen et al., 2023) | 4× | 40.85 | 33.30 | 0.9083 | 29.47 | 0.8015 | 28.09 | 0.7551 | 28.60 | 0.8498 | 33.09 | 0.9335 |
| Hi-IR-L (Ours) | 4× | 39.22 | 33.22 | 0.9103 | 29.49 | 0.8041 | 28.13 | 0.7622 | 28.72 | 0.8514 | 33.13 | 0.9366 |

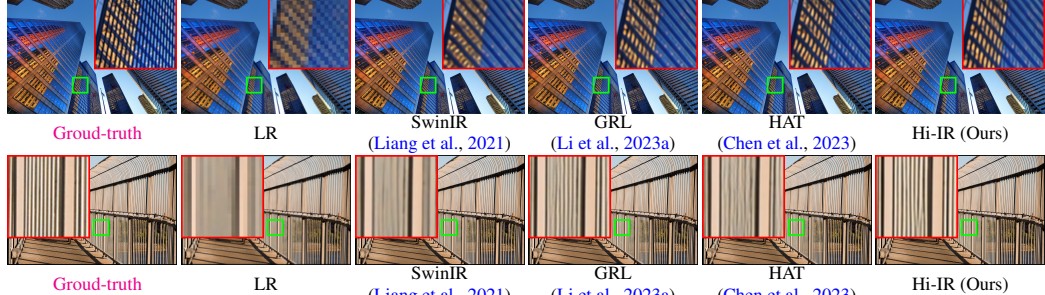

Figure 5: Visual results for classical image ×4 SR on Urban100 dataset.

**Efficiency Analysis.** We report the efficiency comparison results on two IR tasks. For the columnar architecture-based SR, our Hi-IR achieves the best PSNR with much lower parameters (28.6% reduction) and FLOPs (31.1% reduction), and runtime (9.95% reduction) compared to HAT (Chen et al., 2023). Similar observation can also be achieved on the denoising task.

## 5.2 EVALUATION OF HI-IR ON VARIOUS IR TASKS

**Image SR.** For the classical image SR, we compared our Hi-IR with state-of-the-art SR models. The quantitative results are shown in Tab. 8. Aside from the 2nd-best results across all scales on Set5 and the 2nd-best results for the 2× scale on Set14, the proposed Hi-IR archives the best PSNR and SSIM on all other test sets across all scales. In particular, significant improvements in terms of the PSNR on Urban100 (*i.e.,* 0.13 dB for 2× SR of the base model and 0.12 dB for the 4× SR of the large model) and Manga109 (*i.e.,* 0.21 dB for 2× SR) compared to HAT (Chen et al., 2023), but with fewer trainable parameters. The visual results shown in Fig. 5 also validate the effectiveness of the proposed Hi-IR in restoring more details and structural content. More results are in Tab. 19 of Appx. C, Fig. 10 to Fig. 12 of Appx. E.

**Image Denoising.** We provide both the color and the grayscale image denoising results in Tab. 9. Our approach demonstrates superior performance on diverse datasets, including Kodak24, McMaster, and Urban100 for color image denoising, as well as Set12 and Urban100 for grayscale image denoising. These comparative analyses serve to reinforce the efficacy of the proposed Hi-IR, suggesting that it may exhibit a higher degree of generalization. Additionally, a closer examination of more visual results is available in Fig. 13 of Appx. E, further substantiates the capabilities of Hi-IR. These results illustrate its proficiency in effectively eliminating heavy noise corruption while preserving high-frequency image details. The outcome is sharper edges and more natural textures, with no discernible issues of over-smoothness or over-sharpness.

Table 9: *Color and grayscale image denoising* results.

| Method | Params [M] | Color | | | | | | | | | Grayscale | | | | | |
|---|---|---|---|---|---|---|---|---|---|---|---|---|---|---|---|---|
| | | Kodak24 | | | McMaster | | | Urban100 | | | Set12 | | | Urban100 | | |
| | | $\sigma$=15 | $\sigma$=25 | $\sigma$=50 | $\sigma$=15 | $\sigma$=25 | $\sigma$=50 | $\sigma$=15 | $\sigma$=25 | $\sigma$=50 | $\sigma$=15 | $\sigma$=25 | $\sigma$=50 | $\sigma$=15 | $\sigma$=25 | $\sigma$=50 |
| DnCNN (Kiku et al., 2016) | 0.56 | 34.60 | 32.14 | 28.95 | 33.45 | 31.52 | 28.62 | 32.98 | 30.81 | 27.59 | 32.86 | 30.44 | 27.18 | 32.64 | 29.95 | 26.26 |
| RNAN (Zhang et al., 2019) | 8.96 | - | - | 29.58 | - | - | 29.72 | - | - | 29.08 | - | - | 27.70 | - | - | 27.65 |
| IPT (Chen et al., 2021) | 115.33 | - | - | 29.64 | - | - | 29.98 | - | - | 29.71 | - | - | - | - | - | - |
| EDT-B (Li et al., 2021) | 11.48 | 35.37 | 32.94 | 29.87 | 35.61 | 33.34 | 30.25 | 35.22 | 33.07 | 30.16 | - | - | - | - | - | - |
| DRUNet (Zhang et al., 2021) | 32.64 | 35.31 | 32.89 | 29.86 | 35.40 | 33.14 | 30.08 | 34.81 | 32.60 | 29.61 | 33.25 | 30.94 | 27.90 | 33.44 | 31.11 | 27.96 |
| SwinIR (Liang et al., 2021) | 11.75 | 35.34 | 32.89 | 29.79 | 35.61 | 33.20 | 30.22 | 35.13 | 32.90 | 29.82 | 33.36 | 31.01 | 27.91 | 33.70 | 31.30 | 27.98 |
| Restormer (Zamir et al., 2022) | 26.13 | 35.47 | 33.04 | 30.01 | 35.61 | 33.34 | 30.30 | 35.13 | 32.96 | 30.02 | 33.42 | 31.08 | 28.00 | 33.79 | 31.46 | 28.29 |
| Xformer (Zhang et al., 2023a) | 25.23 | 35.39 | 32.99 | 29.94 | 35.68 | 33.44 | 30.38 | 35.29 | 33.21 | 30.36 | 33.46 | 31.16 | 28.10 | 33.98 | 31.78 | 28.71 |
| Hi-IR (Ours) | 22.33 | 35.42 | 33.01 | 29.98 | 35.69 | 33.44 | 30.42 | 35.46 | 33.34 | 30.59 | 33.48 | 31.19 | 28.15 | 34.11 | 31.92 | 28.91 |

Table 10: *Grayscale image JPEG compression artifact removal* results. †A single model is trained to handle multiple noise levels.

| Set | QF | JPEG | | †DnCNN3 | | †DRUNet | | †Hi-IR (Ours) | | SwinIR | | ART | | CAT | | Hi-IR (Ours) | |
|---|---|---|---|---|---|---|---|---|---|---|---|---|---|---|---|---|---|
| | | PSNR↑ | SSIM↑ | PSNR↑ | SSIM↑ | PSNR↑ | SSIM↑ | PSNR↑ | SSIM↑ | PSNR↑ | SSIM↑ | PSNR↑ | SSIM↑ | PSNR↑ | SSIM↑ | PSNR↑ | SSIM↑ |
| Classic5 | 10 | 27.82 | 0.7600 | 29.40 | 0.8030 | 30.16 | 0.8234 | 30.25 | 0.8236 | 30.27 | 0.8249 | 30.27 | 0.8258 | 30.26 | 0.8250 | 30.38 | 0.8266 |
| | 20 | 30.12 | 0.8340 | 31.63 | 0.8610 | 32.39 | 0.8734 | 32.51 | 0.8737 | 32.52 | 0.8748 | - | - | 32.57 | 0.8754 | 32.62 | 0.8751 |
| | 30 | 31.48 | 0.8670 | 32.91 | 0.8860 | 33.59 | 0.8949 | 33.74 | 0.8954 | 33.73 | 0.8961 | 33.74 | 0.8964 | 33.77 | 0.8964 | 33.80 | 0.8962 |
| | 40 | 32.43 | 0.8850 | 33.77 | 0.9000 | 34.41 | 0.9075 | 34.55 | 0.9078 | 34.52 | 0.9082 | 34.55 | 0.9086 | 34.58 | 0.9087 | 34.61 | 0.9082 |
| LIVE1 | 10 | 27.77 | 0.7730 | 29.19 | 0.8120 | 29.79 | 0.8278 | 29.84 | 0.8328 | 29.86 | 0.8287 | 29.89 | 0.8300 | 29.89 | 0.8295 | 29.94 | 0.8359 |
| | 20 | 30.07 | 0.8510 | 31.59 | 0.8800 | 32.17 | 0.8899 | 32.24 | 0.8926 | 32.25 | 0.8909 | - | - | 32.30 | 0.8913 | 32.31 | 0.8938 |
| | 30 | 31.41 | 0.8850 | 32.98 | 0.9090 | 33.59 | 0.9166 | 33.67 | 0.9192 | 33.69 | 0.9174 | 33.71 | 0.9178 | 33.73 | 0.9177 | 33.73 | 0.9223 |
| | 40 | 32.35 | 0.9040 | 33.96 | 0.9250 | 34.58 | 0.9312 | 34.66 | 0.9347 | 34.67 | 0.9317 | 34.70 | 0.9322 | 34.72 | 0.9320 | 34.71 | 0.9347 |
| Urban100 | 10 | 26.33 | 0.7816 | 28.54 | 0.8484 | 30.31 | 0.8745 | 30.62 | 0.8808 | 30.55 | 0.8835 | 30.87 | 0.8894 | 30.81 | 0.8866 | 31.07 | 0.8950 |
| | 20 | 28.57 | 0.8545 | 31.01 | 0.9050 | 32.81 | 0.9241 | 33.21 | 0.9256 | 33.12 | 0.9190 | - | - | 33.38 | 0.9269 | 33.51 | 0.9250 |
| | 30 | 30.00 | 0.9013 | 32.47 | 0.9312 | 34.23 | 0.9414 | 34.64 | 0.9478 | 34.58 | 0.9417 | 34.81 | 0.9442 | 34.81 | 0.9449 | 34.86 | 0.9459 |
| | 40 | 31.06 | 0.9215 | 33.49 | 0.9412 | 35.20 | 0.9547 | 35.63 | 0.9566 | 35.50 | 0.9515 | 35.73 | 0.9553 | 35.73 | 0.9511 | 35.77 | 0.9561 |

Table 11: *Single-image motion deblurring* on GoPro and HIDE dataset. GoPro dataset is used for training.

| Method | GoPro | HIDE | Average |
|---|---|---|---|
| | PSNR↑ / SSIM↑ | PSNR↑ / SSIM↑ | PSNR↑ / SSIM↑ |
| DeblurGAN-v2 (Kupyn et al., 2019) | 29.55 / 0.934 | 26.61 / 0.875 | 28.08 / 0.905 |
| SRN (Tao et al., 2018) | 30.26 / 0.934 | 28.36 / 0.915 | 29.31 / 0.925 |
| SPAIR (Purohit et al., 2021) | 32.06 / 0.953 | 30.29 / 0.931 | 31.18 / 0.942 |
| MIMO-UNet+ (Cho et al., 2021) | 32.45 / 0.957 | 29.99 / 0.930 | 31.22 / 0.944 |
| MPRNet (Zamir et al., 2021) | 32.66 / 0.959 | 30.96 / 0.939 | 31.81 / 0.949 |
| MAXIM-3S (Tu et al., 2022) | 32.86 / 0.961 | 32.83 / 0.956 | 32.85 / 0.959 |
| Restormer (Zamir et al., 2022) | 32.92 / 0.961 | 31.22 / 0.942 | 32.07 / 0.952 |
| Stripformer (Tsai et al., 2022a) | 33.08 / 0.962 | 31.03 / 0.940 | 32.06 / 0.951 |
| ShuffleFormer (Xiao et al., 2023) | 33.38 / 0.965 | 31.25 / 0.943 | 31.32 / 0.954 |
| GRL-B (Li et al., 2023a) | 33.93 / 0.968 | 31.65 / 0.947 | 32.79 / 0.958 |
| Hi-IR-L (Ours) | 33.99 / 0.968 | 31.64 / 0.947 | 32.82 / 0.958 |

Table 12: *Single image motion deblurring* on *RealBlur dataset.* †: Methods trained on Real-Blur.

| Method | RealBlur-R | RealBlur-J | Average |
|---|---|---|---|
| | PSNR↑ / SSIM↑ | PSNR↑ / SSIM↑ | PSNR↑ / SSIM↑ |
| †DeblurGAN-v2 | 36.44 / 0.935 | 29.69 / 0.870 | 33.07 / 0.903 |
| †SRN (Tao et al., 2018) | 38.65 / 0.965 | 31.38 / 0.909 | 35.02 / 0.937 |
| †MPRNet (Zamir et al., 2021) | 39.31 / 0.972 | 31.76 / 0.922 | 35.54 / 0.947 |
| †MIMO-UNet+ (Cho et al., 2021) | - / - | 32.05 / 0.921 | - / - |
| †MAXIM-3S (Tu et al., 2022) | 39.45 / 0.962 | 32.84 / 0.935 | 36.15 / 0.949 |
| †BANet (Tsai et al., 2022b) | 39.55 / 0.971 | 32.00 / 0.923 | 35.78 / 0.947 |
| †MSSNet (Kim et al., 2022) | 39.76 / 0.972 | 32.10 / 0.928 | 35.93 / 0.950 |
| DeepRFT+ (Mao et al., 2023) | 39.84 / 0.972 | 32.19 / 0.931 | 36.02 / 0.952 |
| †Stripformer (Tsai et al., 2022a) | 39.84 / 0.974 | 32.48 / 0.929 | 36.16 / 0.952 |
| †GRL-B (Li et al., 2023a) | 40.20 / 0.974 | 32.82 / 0.932 | 36.51 / 0.953 |
| †Hi-IR-L (Ours) | 40.40 / 0.976 | 32.92 / 0.933 | 36.66 / 0.954 |

**Image JPEG CAR.** For JPEG CAR, the experiments are conducted for grayscale images with four quality factors (*i.e.,* 10, 20, 30, and 40) under two experimental settings (*i.e.,* †, one single model is trained to handle multiple quality factors, and each model for each image quality). We compare Hi-IR with DnCNN3 (Zhang et al., 2017a), DRUNet (Zhang et al., 2021), SwinIR (Liang et al., 2021), ART (Zhang et al., 2022), CAT (Chen et al., 2022b). Specifically, the quantitative results shown in Tab. 10 validate that the proposed Hi-IR outperforms most of the other comparison methods under both settings. Visual comparisons are provided in Fig. 14 of Appx. E to further support the effectiveness of the proposed Hi-IR.

**Single-Image Motion Deblurring.** The results regarding the single-image motion deblurring are shown in Tab. 11 and Tab. 12. For the synthetic datasets, compared with previous stat-of-the-art GRL (Li et al., 2023a), the proposed Hi-IR achieves the best results on the GoPro dataset and the second-best results on HIDE datasets. For the real dataset, our method also achieves the new state-of-the-art performance of 40.40 PSNR on the RealBlur-R dataset and 32.92 PSNR on the RealBlur-J dataset. The visual results are shown in Fig. 16 and Fig. 17 of Appx. E.

**Defocus Deblurring.** We also validate the effectiveness of our Hi-IR for dual-pixel defocus deblurring. The results in Tab. 13 show that Hi-IR outperforms the previous methods for all three scenes. Compared with Restormer on the combined scenes, our Hi-IR achieves a decent performance boost of 0.35 dB for dual-pixel defocus deblurring.

Table 13: ***Defocus deblurring*** results. **D:** dual-pixel defocus deblurring.

| Method | Indoor Scenes | | | | Outdoor Scenes | | | | Combined | | | |
|---|---|---|---|---|---|---|---|---|---|---|---|---|
| | PSNR↑ | SSIM↑ | MAE↓ | LPIPS↓ | PSNR↑ | SSIM↑ | MAE↓ | LPIPS↓ | PSNR↑ | SSIM↑ | MAE↓ | LPIPS↓ |
| DPDNet$_D$ (Abuolaim & Brown, 2020) | 27.48 | 0.849 | 0.029 | 0.189 | 22.90 | 0.726 | 0.052 | 0.255 | 25.13 | 0.786 | 0.041 | 0.223 |
| RDPD$_D$ (Abuolaim et al., 2021) | 28.10 | 0.843 | 0.027 | 0.210 | 22.82 | 0.704 | 0.053 | 0.298 | 25.39 | 0.772 | 0.040 | 0.255 |
| Uformer$_D$ (Wang et al., 2022) | 28.23 | 0.860 | 0.026 | 0.199 | 23.10 | 0.728 | 0.051 | 0.285 | 25.65 | 0.795 | 0.039 | 0.243 |
| IFAN$_D$ (Lee et al., 2021) | 28.66 | 0.868 | 0.025 | 0.172 | 23.46 | 0.743 | 0.049 | 0.240 | 25.99 | 0.804 | 0.037 | 0.207 |
| Restormer$_D$ (Zamir et al., 2022) | 29.48 | 0.895 | 0.023 | 0.134 | 23.97 | 0.773 | 0.047 | 0.175 | 26.66 | 0.833 | 0.035 | 0.155 |
| Hi-IR$_D$-B (Ours) | 29.70 | 0.902 | 0.023 | 0.116 | 24.46 | 0.798 | 0.045 | 0.154 | 27.01 | 0.848 | 0.034 | 0.135 |

Table 14: ***Image demosaicking*** results.

| Datasets | Matlab | DDR | DeepJoint | RLDD | DRUNet | RNAN | GRL-S | Hi-IR (Ours) |
|---|---|---|---|---|---|---|---|---|
| Kodak | 35.78 | 41.11 | 42.00 | 42.49 | 42.68 | 43.16 | 43.57 | 43.69 |
| McMaster | 34.43 | 37.12 | 39.14 | 39.25 | 39.39 | 39.70 | 40.22 | 40.78 |

Table 15: ***IR in AWC*** results.

| Dataset | All-in-One | TransWeather | SemanIR | Ours |
|---|---|---|---|---|
| **RainDrop** | 31.12 | 28.84 | 30.82 | 30.84 |
| **Test1 (rain+fog)** | 24.71 | 27.96 | 29.57 | 30.93 |
| **SnowTest100k-L** | 28.33 | 28.48 | 30.76 | 30.85 |

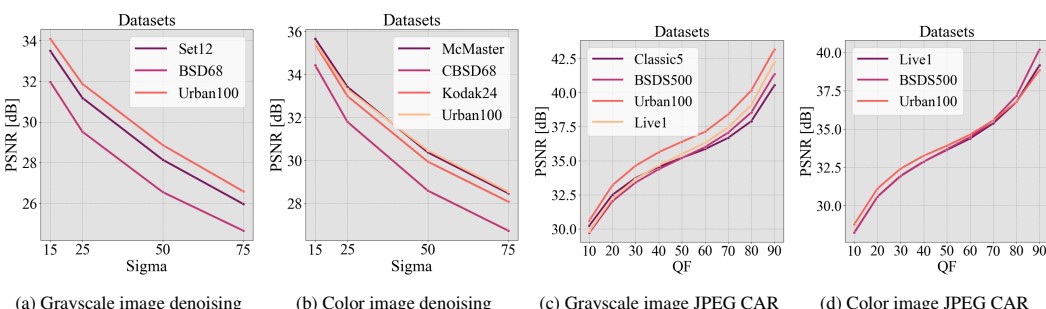

(a) Grayscale image denoising  (b) Color image denoising  (c) Grayscale image JPEG CAR  (d) Color image JPEG CAR

Figure 6: Training one model for multiple degradation levels.

**Image Demosaicking.** We compare DDR (Wu et al., 2016), DeepJoint (Gharbi et al., 2016), RLDD (Guo et al., 2020), DRUNet (Zhang et al., 2021), RNAN (Zhang et al., 2019), and GRL-S (Li et al., 2023a) with the proposed method for demosaicking in Tab. 14. It shows that the proposed Hi-IR archives the best performance on both the Kodak and MaMaster test datasets. Especially, 0.12 dB and 0.56 dB absolute improvement compared to the current state-of-the-art GRL.

**One model for multiple degradation levels.** For image denoising and JPEG CAR, we trained a single model to handle multiple degradation levels. This setup makes it possible to apply one model to deal with images that have been degraded under different conditions, making the model more flexible and generalizable. During training, the noise level is randomly sampled from the range [15, 75] while the JPEG compression quality factor is randomly sampled from the range [10, 90]. The degraded images are generated online. During the test phase, the degradation level is fixed to a certain value. The experimental results are summarized in Fig. 6. The numerical results for grayscale JPEG CAR are presented in Tab. 10. These results show that in the one-model-multiple-degradation setting †, the proposed Hi-IR achieves the best performance.

**IR in AWC.** We validate Hi-IR in adverse weather conditions like rain+fog (Test1 (Li et al., 2020)), snow (SnowTest100K-L (Liu et al., 2018)), and raindrops (RainDrop (Qian et al., 2018)). We compare Hi-IR with All-in-One (Li et al., 2020) TransWeather (Valanarasu et al., 2022), and Se-manIR (Ren et al., 2024). The PSNR score is reported in Tab. 15 for each method. Our method achieves the best performance on Test1 (*i.e.,* 4.6% improvement) and SnowTest100k-L (*i.e.,* 0.09 dB improvement), while the second-best PSNR on RainDrop compared to all other methods. The visual comparison presented in Fig. 15 of Appx. E also shows that our method can restore better structural context and cleaner details.

# 6 CONCLUSION

In this paper, we introduced a hierarchical information flow principle for IR. Leveraging this concept, we devised a new model called Hi-IR, which progressively propagates information within local regions, facilitates information exchange in non-local ranges, and mitigates information isolation in the global context. We investigated how to scale up an IR model. The effectiveness and generalizability of Hi-IR was validated through comprehensive experiments across various IR tasks.

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

# APPENDIX

## A EXPERIMENTAL SETTINGS

### A.1 ARCHITECTURE DETAILS

We choose two commonly used basic architectures for IR tasks including the U-shape hierarchical architecture and the columnar architecture. The columnar architecture is used for image SR while the U-shape architecture is used for other IR tasks including image denoising, JPEG CAR, image deblurring, IR in adverse weather conditions, image deblurring, and image demosaicking. We included details on the structure of the Hi-IR in Tab. 16. This table outlines the number of Hi-IR stages and the distribution of Hi-IR layers within each stage for a thorough understanding of our model's architecture.

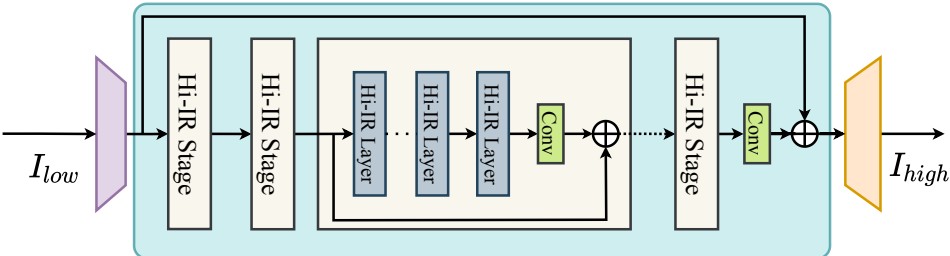

Figure 7: The columnar Hi-IR architecture.

Table 16: The details of the Hi-IR stages and Hi-IR layers per stage for both architectures.

|  | U-shaped architecture | | | Columnar architecture | |
|---|---|---|---|---|---|
|  | Down Stages | Upstages | Latent Stage | Hi-IR-Base | Hi-IR-Large |
| Num. of Hi-IR Stages | 3 | 3 | 1 | 6 | 8 |
| Num. of Hi-IR Layer/Stage | 6 | 6 | 6 | 6 | 8 |

### A.2 TRAINING DETAILS

The proposed Hi-IR explores **7** different IR tasks, and the training settings vary slightly for each task. These differences encompass the architecture of the proposed Hi-IR, variations in training phases, choice of the optimizer, employed loss functions, warm-up settings, learning rate schedules, batch sizes, and patch sizes. We have provided a comprehensive overview of these details.

In addition, there are several points about the training details we want to make further explanation. 1) For image SR, the network is pre-trained on ImageNet (Deng et al., 2009). This is inspired by previous works (Dong et al., 2014; Chen et al., 2021; Li et al., 2021; Chen et al., 2023). 2) The optimizer used for IR in AWC is Adam (Kingma & Ba, 2014), while AdamW (Loshchilov & Hutter, 2018) is used for the rest IR tasks. 3) The training losses for IR in AWC are the smooth L1 and the Perception VGG loss (Johnson et al., 2016; Simonyan & Zisserman, 2015). For image deblurring, the training loss is the Charbonnier loss. For the rest IR task, the L1 loss is commonly used during the training. 4) For IR in AWC, we adopted similar training settings as Transweather (Valanarasu et al., 2022), the model is trained for a total of 750K iterations.

### A.3 DATA AND EVALUATION

The training dataset and test datasets for different IR tasks are described in this section. For IR in AWC, we used a similar training pipeline as Transweather with only one phase. Additionally, for tasks such as image super-resolution (SR), JPEG CAR, image denoising, and demosaicking, how the corresponding low-quality images are generated is also briefly introduced below.

**Image SR.** For image SR, the LR image is synthesized by `Matlab` bicubic downsampling function before the training. We investigated the upscalingg factors $\times 2$, $\times 3$, and $\times 4$.

- The training datasets: DIV2K (Agustsson & Timofte, 2017) and Flickr2K (Lim et al., 2017).

- The test datasets: Set5 (Bevilacqua et al., 2012), Set14 (Zeyde et al., 2010), BSD100 (Martin et al., 2001), Urban100 (Huang et al., 2015), and Manga109 (Matsui et al., 2017).

**Image Denoising.** For image denoising, we conduct experiments on both color and grayscale image denoising. During training and testing, noisy images are generated by adding independent additive white Gaussian noise (AWGN) to the original images. The noise levels are set to $\sigma = 15, 25, 50$. We train individual networks at different noise levels. The network takes the noisy images as input and tries to predict noise-free images. Additionally, we also tried to train one model for all noise levels.

- The training datasets: DIV2K (Agustsson & Timofte, 2017), Flickr2K (Lim et al., 2017), WED (Ma et al., 2016), and BSD400 (Martin et al., 2001).
- The test datasets for color image: CBSD68 (Martin et al., 2001), Kodak24 (Franzen, 1999), McMaster (Zhang et al., 2011), and Urban100 (Huang et al., 2015).
- The test datasets for grayscale image: Set12 (Zhang et al., 2017a), BSD68 (Martin et al., 2001), and Urban100 (Huang et al., 2015).

**JPEG compression artifact removal.** For JPEG compression artifact removal, the JPEG image is compressed by the `cv2` JPEG compression function. The compression function is characterized by the quality factor. We investigated four compression quality factors including 10, 20, 30, and 40. The smaller the quality factor, the more the image is compressed, meaning a lower quality. We also trained one model to deal with different quality factors.

- The training datasets: DIV2K (Agustsson & Timofte, 2017), Flickr2K (Lim et al., 2017), and WED (Ma et al., 2016).
- The test datasets: Classic5 (Foi et al., 2007), LIVE1 (Sheikh, 2005), Urban100 (Huang et al., 2015), BSD500 (Arbelaez et al., 2010).

**IR in Adverse Weather Conditions.** For IR in adverse weather conditions, the model is trained on a combination of images degraded by a variety of adverse weather conditions. The same training and test dataset is used as in Transweather (Valanarasu et al., 2022). The training data comprises 9,000 images sampled from Snow100K (Liu et al., 2018), 1,069 images from Raindrop (Qian et al., 2018), and 9,000 images from Outdoor-Rain (Li et al., 2019a). Snow100K includes synthetic images degraded by snow, Raindrop consists of real raindrop images, and Outdoor-Rain contains synthetic images degraded by both fog and rain streaks. The proposed method is tested on both synthetic and real-world datasets.

- The test datasets: test1 dataset (Li et al., 2020; 2019a), the RainDrop test dataset (Qian et al., 2018), and the Snow100k-L test.

**Image Deblurring.** For single-image motion deblurring,

- The training datasets: GoPro (Nah et al., 2017) dataset.
- The test datasets: GoPro (Nah et al., 2017), HIDE (Shen et al., 2019), RealBlur-R (Rim et al., 2020), and RealBlur-J (Rim et al., 2020) datasets.

**Defocus Deblurring.** The task contains two modes including single-image defocus deblurring and dual-pixel defocus deblurring. For single-image defocus deblurring, only the blurred central-view image is available. For dual-pixel defocus deblurring, both the blurred left-view and right-view images are available. The dual-pixel images could provide additional information for defocus deblurring and thus could lead to better results. PSNR, SSIM, and mean absolute error (MAE) on the RGB channels are reported. Additionally, the image perceptual quality score LPIPS is also reported.

- The training datasets: DPDD (Abuolaim & Brown, 2020) training dataset. The training subset contains 350 scenes.
- The test datasets: DPDD (Abuolaim & Brown, 2020) test dataset. The test set contains 37 indoor scenes and 39 outdoor scenes

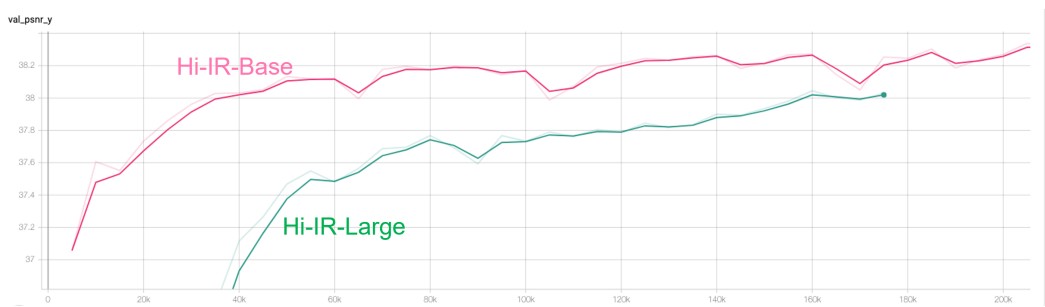

Figure 8: When the SR model is scale-up from Hi-IR-L to Hi-IR-B, the model Hi-IR-L converges slower than Hi-IR-B.

**Image Demosaicking.** For image demosaicking, the mosaic image is generated by applying a Bayer filter on the ground-truth image. Then the network try to restore high-quality image. The mosaic image is first processed by the default `Matlab` demosaic function and then passed to the network as input.

- The training datasets: DIV2K (Agustsson & Timofte, 2017) and Flickr2K (Lim et al., 2017).

- The test datasets: Kodak (Franzen, 1999), McMaster (Zhang et al., 2011).

## B  MODEL SCALING-UP

As mentioned in the main paper, when the initially designed SR model is scaled up from about 10M parameters to about 50M parameters, the performance of the large SR model becomes worse. The effect is shown in Fig. 8. The PSNR curve on the Set5 dataset for the first 200k iterations is shown in this figure. The scale-up model Hi-IR-L converges slower than the smaller model Hi-IR-B. The same phenomenon could be observed by comparing the first two rows for each upscaling factor in Tab. 17, where scaled-up models converge to worse local minima. A similar problem occurs in previous works (Chen et al., 2023; Lim et al., 2017).

Table 17: Model scaling-up exploration with SR.

| Scale | Model Size | Warm up | Conv Type | PSNR | | | | |
|---|---|---|---|---|---|---|---|---|
| | | | | Set5 | Set14 | BSD100 | Urban100 | Manga109 |
| 2× | 15.69 | No | conv1 | 38.52 | 34.47 | 32.56 | 34.17 | 39.77 |
| 2× | 57.60 | No | conv1 | 38.33 | 34.17 | 32.46 | 33.60 | 39.37 |
| 2× | 57.60 | Yes | conv1 | 38.41 | 34.33 | 32.50 | 33.80 | 39.51 |
| 2× | 54.23 | Yes | linear | 38.56 | 34.59 | 32.58 | 34.32 | 39.87 |
| 2× | 55.73 | Yes | conv3 | 38.65 | 34.48 | 32.58 | 34.33 | 40.12 |
| 3× | 15.87 | No | conv1 | 35.06 | 30.91 | 29.48 | 30.02 | 34.41 |
| 3× | 57.78 | No | conv1 | 34.70 | 30.62 | 29.33 | 29.11 | 33.96 |
| 3× | 57.78 | Yes | conv1 | 34.91 | 30.77 | 29.39 | 29.53 | 34.12 |
| 3× | 54.41 | Yes | linear | 35.13 | 31.04 | 29.52 | 30.20 | 34.54 |
| 3× | 55.91 | Yes | conv3 | 35.14 | 31.03 | 29.51 | 30.22 | 34.76 |
| 4× | 15.84 | No | conv1 | 33.00 | 29.11 | 27.94 | 27.67 | 31.41 |
| 4× | 57.74 | No | conv1 | 33.08 | 29.19 | 27.97 | 27.83 | 31.56 |
| 4× | 57.74 | Yes | conv1 | 32.67 | 28.93 | 27.83 | 27.11 | 30.97 |
| 4× | 54.37 | Yes | linear | 33.06 | 29.16 | 27.99 | 27.93 | 31.66 |
| 4× | 55.88 | Yes | conv3 | 33.06 | 29.16 | 27.97 | 27.87 | 31.54 |

### B.1  WHY DOES REPLACING HEAVYWEIGHT CONVOLUTION WORK?

We hypothesize that replacing dense $3 \times 3$ convolutions with linear layers and bottleneck blocks works because of the initialization and backpropagation of the network.

In the Xavier and Kaiming weight initialization method, the magnitude of the weights is inversely related to `fan_in`/`fan_out` of a layer which is the multiplication of the number of input and output

Table 18: Comparison of SR results between dot production attention and cosine similarity attention for scaled-up models.

| Scale | Attn. type | Set5 | Set14 | BSD100 | Urban100 | Manga109 |
|-------|------------|------|-------|--------|----------|----------|
| ×2 | dot prod | 38.56 | 34.79 | 32.63 | 34.49 | 39.89 |
| ×2 | cosine sim | 38.43 | 34.65 | 32.56 | 34.13 | 39.69 |
| ×3 | dot prod | 34.98 | 30.98 | 29.45 | 30.06 | 34.35 |
| ×3 | cosine sim | 34.92 | 30.86 | 29.40 | 29.82 | 34.18 |
| ×4 | dot prod | 33.14 | 29.09 | 27.98 | 27.96 | 31.44 |
| ×4 | cosine sim | 33.08 | 29.15 | 27.96 | 27.90 | 31.40 |

channels and kernel size, namely,

$$f_{in} = c_{in} \times k^2, \tag{2}$$

$$f_{out} = c_{out} \times k^2, \tag{3}$$

where $f_{in}$ and $f_{out}$ denotes `fan_in` and `fan_out`, $c_{in}$ and $c_{out}$ denotes input and output channels, and $k$ is kernel size. Thus, when a dense $3 \times 3$ convolution is used, $f_{in}$ and $f_{out}$ can be large, which leads to small initialized weight parameters. This in turn leads to small gradients during the backpropagation. When the network gets deeper, the vanishing gradients could lead to slow convergence. When dense $3 \times 3$ convolution is replaced by linear layers, the kernel size is reduced to 1. When the bottleneck module is used, the number of input and output channels of the middle $3 \times 3$ convolution in the bottleneck block is also reduced. Thus, both of the two measures decreases the `fan_in` and `fan_out` values, leading to larger initialized weight parameters.

### B.2 WHY DOES WARMUP WORK?

Warmup is effective for training large models primarily because it mitigates issues related to unstable gradients and helps the optimizer gradually adapt to the model's large parameter space (Kalra & Barkeshli, 2024; Goyal, 2017). In the early stages of training, the model's parameters are initialized randomly. A high learning rate at this stage can cause large updates, leading to unstable or divergent training due to exploding or vanishing gradients. Warmup starts with a small learning rate and gradually increases it, allowing the optimizer to find a stable path in the loss landscape before applying larger updates. Warmup enables the model to adapt gradually, avoiding overshooting minima and ensuring smoother convergence.

### B.3 WHY DOES DOT PRODUCT WORK BETTER THAN COSINE SIMILARITY?

As shown in Tab. 18, dot product attention works better than cosine similarity attention. We analyze the gradient of dot product and cosine similary as follows. Suppose $\mathbf{q}$ denotes the query and $\mathbf{k}$ denotes the keys. Then dot product and cosine similarity between $\mathbf{q}$ and $\mathbf{k}$ are denoted as $\mathrm{dot\_prod}(\mathbf{q}, \mathbf{k})$ and $\mathrm{cos\_sim}(\mathbf{q}, \mathbf{k})$. The gradient of dot product with respect to $\mathbf{q}$ is

$$\frac{\partial}{\partial \mathbf{q}} \mathrm{dot\_prod}(\mathbf{q}, \mathbf{k}) = \mathbf{k}. \tag{4}$$

The gradient of cosine similarity with respect to $\mathbf{q}$ is

$$\frac{\partial}{\partial \mathbf{q}} \mathrm{cos\_sim}(\mathbf{q}, \mathbf{k}) = \frac{\mathbf{k}}{\|\mathbf{q}\|\|\mathbf{k}\|} - \frac{(\mathbf{q} \cdot \mathbf{k})\mathbf{q}}{\|\mathbf{q}\|^3\|\mathbf{k}\|} = \frac{1}{\|\mathbf{q}\|} \left( \hat{\mathbf{k}} - \mathrm{cos\_sim}(\mathbf{q}, \mathbf{k})\hat{\mathbf{q}} \right), \tag{5}$$

where $\hat{\mathbf{q}}$ and $\hat{\mathbf{k}}$ are normalized $\mathbf{q}$ and $\mathbf{k}$. The gradients with respect to $\mathbf{k}$ have the similar form. The gradient of cosine similarity involves more terms compared to the gradient of the dot product. This increased complexity in the gradient of cosine similarity makes it more prone to producing large or even unstable gradient values. We conducted a numerical analysis of the gradient values for the two attention methods, with the results presented in Fig. 9. As shown in the figure, the gradient of cosine similarity is indeed more prone to producing large values. This issue becomes more pronounced as the model scales up.

## C MORE QUANTITATIVE EXPERIMENTAL RESULTS

Due to the limited space in the main manuscript, we only report a part of the experimental result. In this section, we show the full quantitative experimental results for each IR task in the following.

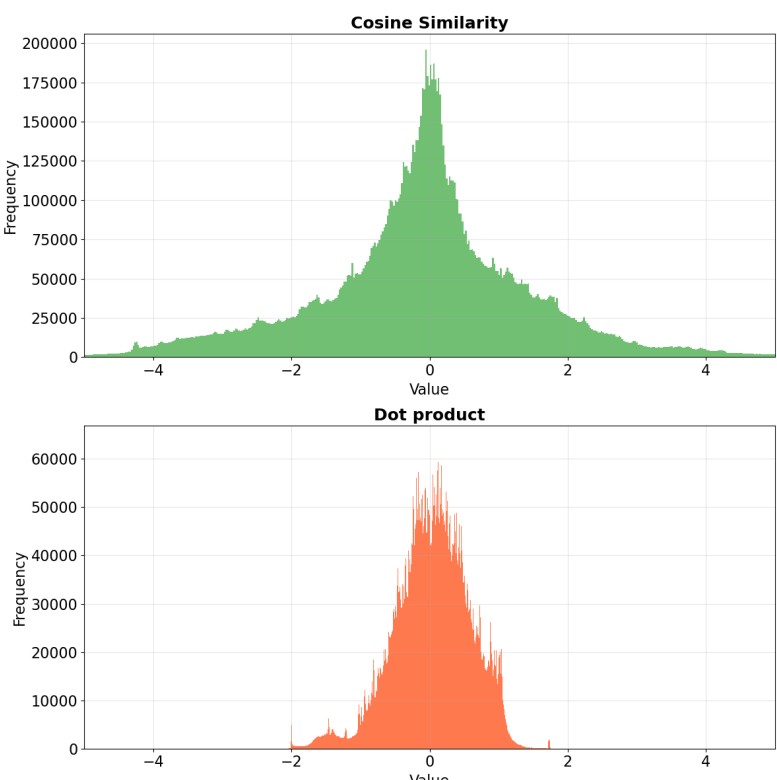

Figure 9: Comparsion of gradients between dot product and cosine similarity.

## C.1 FULL RESULTS ON IMAGE SR

The full numerical comparison results among the proposed Hi-IR and other state-of-the-art methods are presented in Tab. 19. It shows that besides the second-best results on Set5 and the ×2 on Set14, our method achieves all the best results across various scale factors and datasets. Especially, we outperform HAT (Chen et al., 2023) with less trainable parameters.

## C.2 RESULTS FOR COLOR IMAGE JPEG COMPRESSION ARTIFACT REMOVAL

The following methods are compared for color image JPEG artifact removal including QGAC (Ehrlich et al., 2020), FBCNN (Jiang et al., 2021), DRUNet (Zhang et al., 2021), SwinIR (Liang et al., 2021), GRL-S (Li et al., 2023a), and Hi-IR. The results shown in Tab. 20 also validate the effectiveness of the proposed Hi-IR.

## C.3 SINGLE-IMAGE DEFOCUS DEBLURRING

In addition to the dual-pixel defocus deblurring results, we also shown single-image defocus deblurring results in Tab. 21

## C.4 GENERALIZING ONE MODEL TO MORE TYPES DEGRADATIONS

To validate the generalization capability of the proposed method to different types of degradation, we conducted the following experiments. First, we used the same model for both denoising and JPEG compression artifact removal tasks. Notably, a single model was trained to handle varying levels of degradation. The experimental results for denoising are shown in Tab. 22 while the results for JPEG compression artifact removal are shown in Tab. 20 and Tab. 10. Second, we performed experiments on image restoration under adverse weather conditions, including rain, fog, and snow. The results are shown in Tab. 15. Third, we further investigated a one-in-all image restoration setup, encompass-

Table 19: ***Classical image SR*** results. Top-2 results are highlighted in red and blue.

| Method | Scale | Params [M] | Set5 PSNR↑ | Set5 SSIM↑ | Set14 PSNR↑ | Set14 SSIM↑ | BSD100 PSNR↑ | BSD100 SSIM↑ | Urban100 PSNR↑ | Urban100 SSIM↑ | Manga109 PSNR↑ | Manga109 SSIM↑ |
|---|---|---|---|---|---|---|---|---|---|---|---|---|
| EDSR (Lim et al., 2017) | ×2 | 40.73 | 38.11 | 0.9602 | 33.92 | 0.9195 | 32.32 | 0.9013 | 32.93 | 0.9351 | 39.10 | 0.9773 |
| SRFBN (Li et al., 2019b) | ×2 | 2.14 | 38.11 | 0.9609 | 33.82 | 0.9196 | 32.29 | 0.9010 | 32.62 | 0.9328 | 39.08 | 0.9779 |
| RCAN (Zhang et al., 2018b) | ×2 | 15.44 | 38.27 | 0.9614 | 34.12 | 0.9216 | 32.41 | 0.9027 | 33.34 | 0.9384 | 39.44 | 0.9786 |
| SAN (Dai et al., 2019) | ×2 | 15.71 | 38.31 | 0.9620 | 34.07 | 0.9213 | 32.42 | 0.9028 | 33.10 | 0.9370 | 39.32 | 0.9792 |
| HAN (Niu et al., 2020) | ×2 | 63.61 | 38.27 | 0.9614 | 34.16 | 0.9217 | 32.41 | 0.9027 | 33.35 | 0.9385 | 39.46 | 0.9785 |
| NLSA (Mei et al., 2021) | ×2 | 42.63 | 38.34 | 0.9618 | 34.08 | 0.9231 | 32.43 | 0.9027 | 33.42 | 0.9394 | 39.59 | 0.9789 |
| IPT (Chen et al., 2021) | ×2 | 115.48 | 38.37 | - | 34.43 | - | 32.48 | - | 33.76 | - | - | - |
| SwinIR (Liang et al., 2021) | ×2 | 11.75 | 38.42 | 0.9623 | 34.46 | 0.9250 | 32.53 | 0.9041 | 33.81 | 0.9427 | 39.92 | 0.9797 |
| CAT-A (Chen et al., 2022b) | ×2 | 16.46 | 38.51 | 0.9626 | 34.78 | 0.9265 | 32.59 | 0.9047 | 34.26 | 0.9440 | 40.10 | 0.9805 |
| ART (Zhang et al., 2022) | ×2 | 16.40 | 38.56 | 0.9629 | 34.59 | 0.9267 | 32.58 | 0.9048 | 34.3 | 0.9452 | 40.24 | 0.9808 |
| EDT (Li et al., 2021) | ×2 | 11.48 | 38.63 | 0.9632 | 34.80 | 0.9273 | 32.62 | 0.9052 | 34.27 | 0.9456 | 40.37 | 0.9811 |
| GRL-B (Li et al., 2023a) | ×2 | 20.05 | 38.67 | 0.9647 | 35.08 | 0.9303 | 32.67 | 0.9087 | 35.06 | 0.9505 | 40.67 | 0.9818 |
| HAT (Chen et al., 2023) | ×2 | 20.62 | 38.73 | 0.9637 | 35.13 | 0.9282 | 32.69 | 0.9060 | 34.81 | 0.9489 | 40.71 | 0.9819 |
| Hi-IR-B (Ours) | ×2 | 14.68 | 38.71 | 0.9657 | 35.16 | 0.9299 | 32.73 | 0.9087 | 34.94 | 0.9484 | 40.81 | 0.9830 |
| HAT-L (Chen et al., 2023) | ×2 | 40.70 | 38.91 | 0.9646 | 35.29 | 0.9293 | 32.74 | 0.9066 | 35.09 | 0.9505 | 41.01 | 0.9831 |
| Hi-IR-L (Ours) | ×2 | 39.07 | 38.87 | 0.9663 | 35.27 | 0.9311 | 32.77 | 0.9092 | 35.16 | 0.9505 | 41.22 | 0.9846 |
| EDSR (Lim et al., 2017) | ×3 | 43.68 | 34.65 | 0.9280 | 30.52 | 0.8462 | 29.25 | 0.8093 | 28.80 | 0.8653 | 34.17 | 0.9476 |
| SRFBN (Li et al., 2019b) | ×3 | 2.83 | 34.70 | 0.9292 | 30.51 | 0.8461 | 29.24 | 0.8084 | 28.73 | 0.8641 | 34.18 | 0.9481 |
| RCAN (Zhang et al., 2018b) | ×3 | 15.63 | 34.74 | 0.9299 | 30.65 | 0.8482 | 29.32 | 0.8111 | 29.09 | 0.8702 | 34.44 | 0.9499 |
| SAN (Dai et al., 2019) | ×3 | 15.90 | 34.75 | 0.9300 | 30.59 | 0.8476 | 29.33 | 0.8112 | 28.93 | 0.8671 | 34.30 | 0.9494 |
| HAN (Niu et al., 2020) | ×3 | 64.35 | 34.75 | 0.9299 | 30.67 | 0.8483 | 29.32 | 0.8110 | 29.10 | 0.8705 | 34.48 | 0.9500 |
| NLSA (Mei et al., 2021) | ×3 | 45.58 | 34.85 | 0.9306 | 30.70 | 0.8485 | 29.34 | 0.8117 | 29.25 | 0.8726 | 34.57 | 0.9508 |
| IPT (Chen et al., 2021) | ×3 | 115.67 | 34.81 | - | 30.85 | - | 29.38 | - | 29.49 | - | - | - |
| SwinIR (Liang et al., 2021) | ×3 | 11.94 | 34.97 | 0.9318 | 30.93 | 0.8534 | 29.46 | 0.8145 | 29.75 | 0.8826 | 35.12 | 0.9537 |
| CAT-A (Chen et al., 2022b) | ×3 | 16.64 | 35.06 | 0.9326 | 31.04 | 0.8538 | 29.52 | 0.8160 | 30.12 | 0.8862 | 35.38 | 0.9546 |
| ART (Zhang et al., 2022) | ×3 | 16.58 | 35.07 | 0.9325 | 31.02 | 0.8541 | 29.51 | 0.8159 | 30.1 | 0.8871 | 35.39 | 0.9548 |
| EDT (Li et al., 2021) | ×3 | 11.66 | 35.13 | 0.9328 | 31.09 | 0.8553 | 29.53 | 0.8165 | 30.07 | 0.8863 | 35.47 | 0.9550 |
| GRL-B (Li et al., 2023a) | ×3 | 20.24 | 35.12 | 0.9353 | 31.27 | 0.8611 | 29.56 | 0.8235 | 30.92 | 0.8990 | 35.76 | 0.9566 |
| HAT (Chen et al., 2023) | ×3 | 20.81 | 35.16 | 0.9335 | 31.33 | 0.8576 | 29.59 | 0.8177 | 30.7 | 0.8949 | 35.84 | 0.9567 |
| Hi-IR-B (Ours) | ×3 | 14.87 | 35.11 | 0.9372 | 31.37 | 0.8598 | 29.60 | 0.8240 | 30.79 | 0.8977 | 35.92 | 0.9583 |
| HAT-L (Chen et al., 2023) | ×3 | 40.88 | 35.28 | 0.9345 | 31.47 | 0.8584 | 29.63 | 0.8191 | 30.92 | 0.8981 | 36.02 | 0.9576 |
| Hi-IR-L (Ours) | ×3 | 39.26 | 35.20 | 0.9380 | 31.55 | 0.8616 | 29.67 | 0.8256 | 31.07 | 0.9020 | 36.12 | 0.9588 |
| EDSR (Lim et al., 2017) | ×4 | 43.09 | 32.46 | 0.8968 | 28.80 | 0.7876 | 27.71 | 0.7420 | 26.64 | 0.8033 | 31.02 | 0.9148 |
| SRFBN (Li et al., 2019b) | ×4 | 3.63 | 32.47 | 0.8983 | 28.81 | 0.7868 | 27.72 | 0.7409 | 26.60 | 0.8015 | 31.15 | 0.9160 |
| RCAN (Zhang et al., 2018b) | ×4 | 15.59 | 32.63 | 0.9002 | 28.87 | 0.7889 | 27.77 | 0.7436 | 26.82 | 0.8087 | 31.22 | 0.9173 |
| SAN (Dai et al., 2019) | ×4 | 15.86 | 32.64 | 0.9003 | 28.92 | 0.7888 | 27.78 | 0.7436 | 26.79 | 0.8068 | 31.18 | 0.9169 |
| HAN (Niu et al., 2020) | ×4 | 64.20 | 32.64 | 0.9002 | 28.90 | 0.7890 | 27.80 | 0.7442 | 26.85 | 0.8094 | 31.42 | 0.9177 |
| NLSA (Mei et al., 2021) | ×4 | 44.99 | 32.59 | 0.9000 | 28.87 | 0.7891 | 27.78 | 0.7444 | 26.96 | 0.8109 | 31.27 | 0.9184 |
| IPT (Chen et al., 2021) | ×4 | 115.63 | 32.64 | - | 29.01 | - | 27.82 | - | 27.26 | - | - | - |
| SwinIR (Liang et al., 2021) | ×4 | 11.90 | 32.92 | 0.9044 | 29.09 | 0.7950 | 27.92 | 0.7489 | 27.45 | 0.8254 | 32.03 | 0.9260 |
| CAT-A (Chen et al., 2022b) | ×4 | 16.60 | 33.08 | 0.9052 | 29.18 | 0.7960 | 27.99 | 0.7510 | 27.89 | 0.8339 | 32.39 | 0.9285 |
| ART (Zhang et al., 2022) | ×4 | 16.55 | 33.04 | 0.9051 | 29.16 | 0.7958 | 27.97 | 0.751 | 27.77 | 0.8321 | 32.31 | 0.9283 |
| EDT (Li et al., 2021) | ×4 | 11.63 | 33.06 | 0.9055 | 29.23 | 0.7971 | 27.99 | 0.7510 | 27.75 | 0.8317 | 32.39 | 0.9283 |
| GRL-B (Li et al., 2023a) | ×4 | 20.20 | 33.10 | 0.9094 | 29.37 | 0.8058 | 28.01 | 0.7611 | 28.53 | 0.8504 | 32.77 | 0.9325 |
| HAT (Chen et al., 2023) | ×4 | 20.77 | 33.18 | 0.9073 | 29.38 | 0.8001 | 28.05 | 0.7534 | 28.37 | 0.8447 | 32.87 | 0.9319 |
| Hi-IR-B (Ours) | ×4 | 14.83 | 33.14 | 0.9095 | 29.40 | 0.8029 | 28.08 | 0.7611 | 28.44 | 0.8448 | 32.90 | 0.9323 |
| HAT-L (Chen et al., 2023) | ×4 | 40.85 | 33.30 | 0.9083 | 29.47 | 0.8015 | 28.09 | 0.7551 | 28.60 | 0.8498 | 33.09 | 0.9335 |
| Hi-IR-L (Ours) | ×4 | 39.22 | 33.22 | 0.9103 | 29.49 | 0.8041 | 28.13 | 0.7622 | 28.72 | 0.8514 | 33.13 | 0.9366 |

Table 20: ***Color image JPEG compression artifact removal*** *results.*

| Set | QF | JPEG PSNR | JPEG SSIM | †QGAC PSNR | †QGAC SSIM | †FBCNN PSNR | †FBCNN SSIM | †DRUNet PSNR | †DRUNet SSIM | †Hi-IR (Ours) PSNR | †Hi-IR (Ours) SSIM | SwinIR PSNR | SwinIR SSIM | GRL-S PSNR | GRL-S SSIM | Hi-IR (Ours) PSNR | Hi-IR (Ours) SSIM |
|---|---|---|---|---|---|---|---|---|---|---|---|---|---|---|---|---|---|
| LIVE1 | 10 | 25.69 | 0.7430 | 27.62 | 0.8040 | 27.77 | 0.8030 | 27.47 | 0.8045 | 28.24 | 0.8149 | 28.06 | 0.8129 | 28.13 | 0.8139 | 28.36 | 0.8180 |
| LIVE1 | 20 | 28.06 | 0.8260 | 29.88 | 0.8680 | 30.11 | 0.8680 | 30.29 | 0.8743 | 30.59 | 0.8786 | 30.44 | 0.8768 | 30.49 | 0.8776 | 30.66 | 0.8797 |
| LIVE1 | 30 | 29.37 | 0.8610 | 31.17 | 0.8960 | 31.43 | 0.8970 | 31.64 | 0.9020 | 31.95 | 0.9055 | 31.81 | 0.9040 | 31.85 | 0.9045 | 32.02 | 0.9063 |
| LIVE1 | 40 | 30.28 | 0.8820 | 32.05 | 0.9120 | 32.34 | 0.9130 | 32.56 | 0.9174 | 32.88 | 0.9205 | 32.75 | 0.9193 | 32.79 | 0.9195 | 32.94 | 0.9210 |
| BSD500 | 10 | 25.84 | 0.7410 | 27.74 | 0.8020 | 27.85 | 0.7990 | 27.62 | 0.8001 | 28.26 | 0.8070 | 28.22 | 0.8075 | 28.26 | 0.8083 | 28.35 | 0.8092 |
| BSD500 | 20 | 28.21 | 0.8270 | 30.01 | 0.8690 | 30.14 | 0.8670 | 30.39 | 0.8711 | 30.58 | 0.8741 | 30.54 | 0.8739 | 30.57 | 0.8746 | 30.61 | 0.8740 |
| BSD500 | 30 | 29.57 | 0.8650 | 31.330 | 0.8980 | 31.45 | 0.8970 | 31.73 | 0.9003 | 31.93 | 0.9029 | 31.90 | 0.9025 | 31.92 | 0.9030 | 31.99 | 0.9035 |
| BSD500 | 40 | 30.52 | 0.8870 | 32.25 | 0.9150 | 32.36 | 0.9130 | 32.66 | 0.9168 | 32.87 | 0.9193 | 32.84 | 0.9189 | 32.86 | 0.9192 | 32.92 | 0.9195 |
| Urban100 | 10 | 24.46 | 0.7612 | - | - | - | - | 27.10 | 0.8400 | 28.78 | 0.8666 | 28.18 | 0.8586 | 28.54 | 0.8635 | 29.11 | 0.8727 |
| Urban100 | 20 | 26.63 | 0.8310 | - | - | - | - | 30.17 | 0.8991 | 31.12 | 0.9087 | 30.53 | 0.9030 | 30.93 | 0.9067 | 31.36 | 0.9115 |
| Urban100 | 30 | 27.96 | 0.8640 | - | - | - | - | 31.49 | 0.9189 | 32.42 | 0.9265 | 31.87 | 0.9219 | 32.24 | 0.9247 | 32.57 | 0.9279 |
| Urban100 | 40 | 28.93 | 0.8825 | - | - | - | - | 32.36 | 0.9301 | 33.26 | 0.9363 | 32.75 | 0.9329 | 33.09 | 0.9348 | 33.37 | 0.9373 |

Table 21: ***Sinlge-image Defocus deblurring*** results. **S:** single-image defocus deblurring.

| Method | Indoor Scenes PSNR↑ | Indoor Scenes SSIM↑ | Indoor Scenes MAE↓ | Indoor Scenes LPIPS↓ | Outdoor Scenes PSNR↑ | Outdoor Scenes SSIM↑ | Outdoor Scenes MAE↓ | Outdoor Scenes LPIPS↓ | Combined PSNR↑ | Combined SSIM↑ | Combined MAE↓ | Combined LPIPS↓ |
|---|---|---|---|---|---|---|---|---|---|---|---|---|
| EBDB_S (Karaali & Jung, 2017) | 25.77 | 0.772 | 0.040 | 0.297 | 21.25 | 0.599 | 0.058 | 0.373 | 23.45 | 0.683 | 0.049 | 0.336 |
| DMENet_S (Lee et al., 2019) | 25.50 | 0.788 | 0.038 | 0.298 | 21.43 | 0.644 | 0.063 | 0.397 | 23.41 | 0.714 | 0.051 | 0.349 |
| JNB_S (Shi et al., 2015) | 26.73 | 0.828 | 0.031 | 0.273 | 21.10 | 0.608 | 0.064 | 0.355 | 23.84 | 0.715 | 0.048 | 0.315 |
| DPDNet_S (Abuolaim & Brown, 2020) | 26.54 | 0.816 | 0.031 | 0.239 | 22.25 | 0.682 | 0.056 | 0.313 | 24.34 | 0.747 | 0.044 | 0.277 |
| KPAC_S (Son et al., 2021) | 27.97 | 0.852 | 0.026 | 0.182 | 22.62 | 0.701 | 0.053 | 0.269 | 25.22 | 0.774 | 0.040 | 0.227 |
| IFAN_S (Lee et al., 2021) | 28.11 | 0.861 | 0.026 | 0.179 | 22.76 | 0.720 | 0.052 | 0.254 | 25.37 | 0.789 | 0.039 | 0.217 |
| Restormer_S (Zamir et al., 2022) | 28.87 | 0.882 | 0.025 | 0.145 | 23.24 | 0.743 | 0.050 | 0.209 | 25.98 | 0.811 | 0.038 | 0.178 |
| Hi-IR_S-B (Ours) | 28.73 | 0.885 | 0.025 | 0.140 | 23.66 | 0.766 | 0.048 | 0.196 | 26.13 | 0.824 | 0.037 | 0.169 |

ing five different tasks with real-world images. The experimental results in Tab. 23 demonstrate that the proposed method outperforms previous methods by a significant margin. These three sets of ex-

Table 22: ***Color and grayscale image denoising*** results. A single model is trained to handle multiple noise levels.

| Method | Params [M] | Color | | | | Grayscale | |
|---|---|---|---|---|---|---|---|
| | | CBSD68 | Kodak24 | McMaster | Urban100 | Set12 | Urban100 |
| | | $\sigma$=15 $\sigma$=25 $\sigma$=50 | $\sigma$=15 $\sigma$=25 $\sigma$=50 | $\sigma$=15 $\sigma$=25 $\sigma$=50 | $\sigma$=15 $\sigma$=25 $\sigma$=50 | $\sigma$=15 $\sigma$=25 $\sigma$=50 | $\sigma$=15 $\sigma$=25 $\sigma$=50 |
| DnCNN (Kiku et al., 2016) | 0.56 | 33.90 31.24 27.95 | 34.60 32.14 28.95 | 33.45 31.52 28.62 | 32.98 30.81 27.59 | 32.67 30.35 27.18 | 32.28 29.80 26.35 |
| FFDNet (Zhang et al., 2018a) | 0.49 | 33.87 31.21 27.96 | 34.63 32.13 28.98 | 34.66 32.35 29.18 | 33.83 31.40 28.05 | 32.75 30.43 27.32 | 32.40 29.90 26.50 |
| IRCNN (Zhang et al., 2017b) | 0.19 | 33.86 31.16 27.86 | 34.69 32.18 28.93 | 34.58 32.18 28.91 | 33.78 31.20 27.70 | 32.76 30.37 27.12 | 32.46 29.80 26.22 |
| DRUNet (Zhang et al., 2021) | 32.64 | 34.30 31.69 28.51 | 35.31 32.89 29.86 | 35.40 33.14 30.08 | 34.81 32.60 29.61 | 33.25 30.94 27.90 | 33.44 31.11 27.96 |
| Restormer (Zamir et al., 2022) | 26.13 | 34.39 31.78 28.59 | 35.44 33.02 30.00 | 35.55 33.31 30.29 | 35.06 32.91 30.02 | 33.35 31.04 28.01 | 33.67 31.39 28.33 |
| TreeIR (Ours) | 22.33 | 34.43 31.80 28.60 | 35.42 33.00 29.95 | 35.67 33.43 30.38 | 35.46 33.32 30.47 | 33.49 31.18 28.14 | 34.09 31.87 28.86 |

Table 23: **Comparison to state-of-the-art on five degradations.** PSNR ($\uparrow$) and SSIM ($\uparrow$) metrics are reported on the full RGB images with ($\ddagger$) denoting general image restorers, others are specialized all-in-one approaches. Best and second best performances are highlighted.

| Method | Params. | Dehazing SOTS | Deraining Rain100L | Denoising BSD68$_{\sigma=25}$ | Deblurring GoPro | Low-Light LOLv1 | Average |
|---|---|---|---|---|---|---|---|
| NAFNet$\ddagger$ (Chen et al., 2022a) | 17M | 25.23 0.939 | 35.56 0.967 | 31.02 0.883 | 26.53 0.808 | 20.49 0.809 | 27.76 0.881 |
| DGUNet$\ddagger$ (Mou et al., 2022) | 17M | 24.78 0.940 | 36.62 0.971 | 31.10 0.883 | 27.25 0.837 | 21.87 0.823 | 28.32 0.891 |
| SwinIR$\ddagger$ (Liang et al., 2021) | 1M | 21.50 0.891 | 30.78 0.923 | 30.59 0.868 | 24.52 0.773 | 17.81 0.723 | 25.04 0.835 |
| Restormer$\ddagger$ (Zamir et al., 2022) | 26M | 24.09 0.927 | 34.81 0.962 | 31.49 0.884 | 27.22 0.829 | 20.41 0.806 | 27.60 0.881 |
| MambaIR$\ddagger$ (Guo et al., 2024) | 27M | 25.81 0.944 | 36.55 0.971 | 31.41 0.884 | 28.61 0.875 | 22.49 0.832 | 28.97 0.901 |
| DL (Fan et al., 2019) | 2M | 20.54 0.826 | 21.96 0.762 | 23.09 0.745 | 19.86 0.672 | 19.83 0.712 | 21.05 0.743 |
| Transweather (Valanarasu et al., 2022) | 38M | 21.32 0.885 | 29.43 0.905 | 29.00 0.841 | 25.12 0.757 | 21.21 0.792 | 25.22 0.836 |
| TAPE (Liu et al., 2022a) | 1M | 22.16 0.861 | 29.67 0.904 | 30.18 0.855 | 24.47 0.763 | 18.97 0.621 | 25.09 0.801 |
| AirNet (Li et al., 2022a) | 9M | 21.04 0.884 | 32.98 0.951 | 30.91 0.882 | 24.35 0.781 | 18.18 0.735 | 25.49 0.847 |
| IDR (Zhang et al., 2023b) | 15M | 25.24 0.943 | 35.63 0.965 | 31.60 0.887 | 27.87 0.846 | 21.34 0.826 | 28.34 0.893 |
| PromptIR (Potlapalli et al., 2024) | 36M | 26.54 0.949 | 36.37 0.970 | 31.47 0.886 | 28.71 0.881 | 22.68 0.832 | 29.15 0.904 |
| AdaIR (Cui et al., 2024) | 29M | 30.53 0.978 | 38.02 0.981 | 31.35 0.889 | 28.12 0.858 | 23.00 0.845 | 30.20 0.910 |
| Hi-IR (Ours) | 22M | 31.42 0.989 | 38.67 0.985 | 31.58 0.890 | 28.95 0.889 | 23.12 0.851 | 30.75 0.921 |

periments collectively highlight that the proposed hierarchical information flow mechanism enables training a single model that generalizes effectively to various types and levels of degradation.

# D   COMPARISON WITH SHUFFLEFORMER AND SHUFFLE TRANSFORMER

We compare with Random shuffle transformer (ShuffleFormer) (Xiao et al., 2023) and Shuffle transformer (Huang et al., 2021). Both methods use spatial shuffle operations to facilitate non-local information exchange, with one being random and the other deterministic.

Random Shuffle Transformer (ShuffleFormer) (Xiao et al., 2023) applies random shuffling on the spatial dimension, which increases the probability of global information existing within a local window. While this operation extends the receptive field globally in a single step, it compromises the relevance of pixels within the window. In contrast, the hierarchical information flow proposed in this paper progressively propagates information from local to global while preserving the relevance of attended pixels. A comparison with ShuffleFormer on image deblurring is presented in Tab. 11. Hi-IR outperforms ShuffleFormer by a significant margin while using 55.5% fewer parameters. This demonstrates the effectiveness of the hierarchical information flow method introduced in this work.

Shuffle Transformer (Huang et al., 2021) employs a spatial shuffle operation to aggregate information from distant pixels or tokens. However, it differs from the proposed Hi-IR in several key aspects. First, Shuffle Transformer does not enable progressive information propagation within a hierarchical tree structure. Second, its shuffle operation is based on a fixed grid size of $g = 8$. The distance between pixels in the shuffled window is $H/g$ and $W/g$ along the two axes, which directly depends on the image size. For large images (e.g., 1024 pixels), this design forces distant pixels to attend to one another, often introducing irrelevant information. Consequently, this operation is unsuitable for image restoration tasks, where image sizes can become extremely large. In contrast, the L2 information flow attention proposed in this paper limits the maximum patch size, thereby constraining the maximum distance between pixels at this stage. This restriction enhances the relevance of pixel interactions, making it more effective for image restoration tasks.

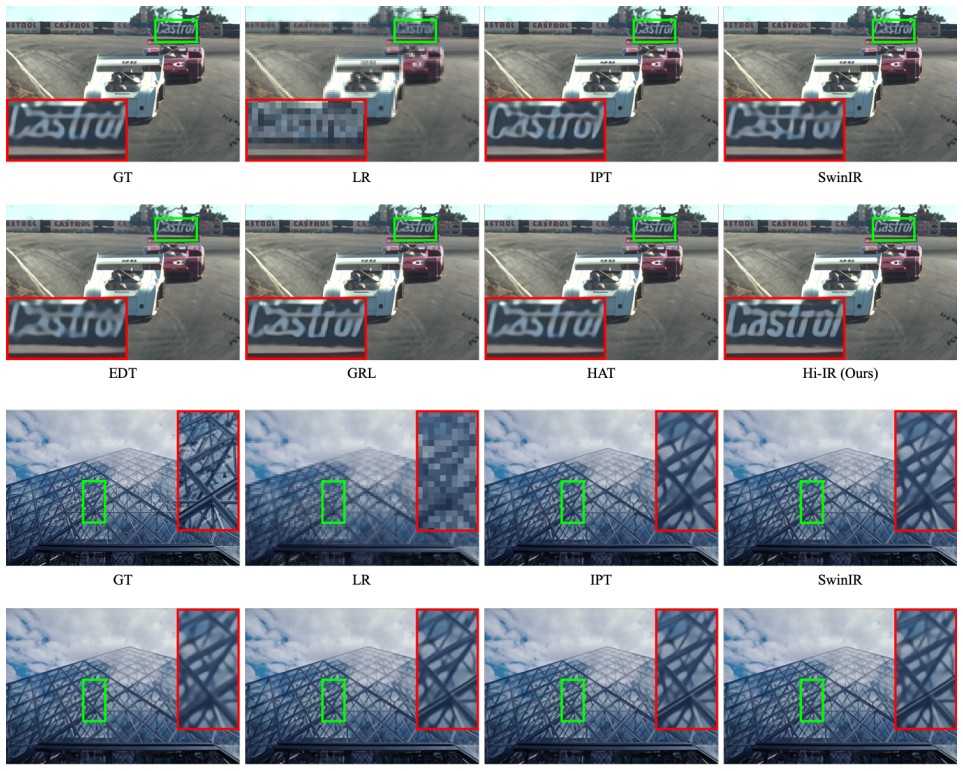

Figure 10: Visual results for classical image ×4 SR on B100 dataset.

# E   MORE VISUAL RESULTS

To further support the effectiveness and generalizability of the proposed Hi-IR intuitively. We provide more visual comparison in terms of image SR (Fig. 10, Fig. 11, and Fig. 12), image denoising (Fig. 13), JPEG compression artifact removal (Fig. 14 ), image restoration in adverse weather conditions(Fig. 15), and single-image deblurring (Fig. 16 and Fig. 17) blow. As shown in those figures, the visual results of the proposed Hi-IR are improved compared with the other methods.

# F   LIMITATIONS

Despite the state-of-the-art performance of Hi-IR, our explorations towards scaling up the model for IR in this paper are still incomplete. Scaling up the IR model is intricate, involving considerations like model design, data collection, and computing resources. We hope our work can catalyze positive impacts on future research, encouraging more comprehensive scaling-up explorations and propelling IR into the domain of large-scale models.

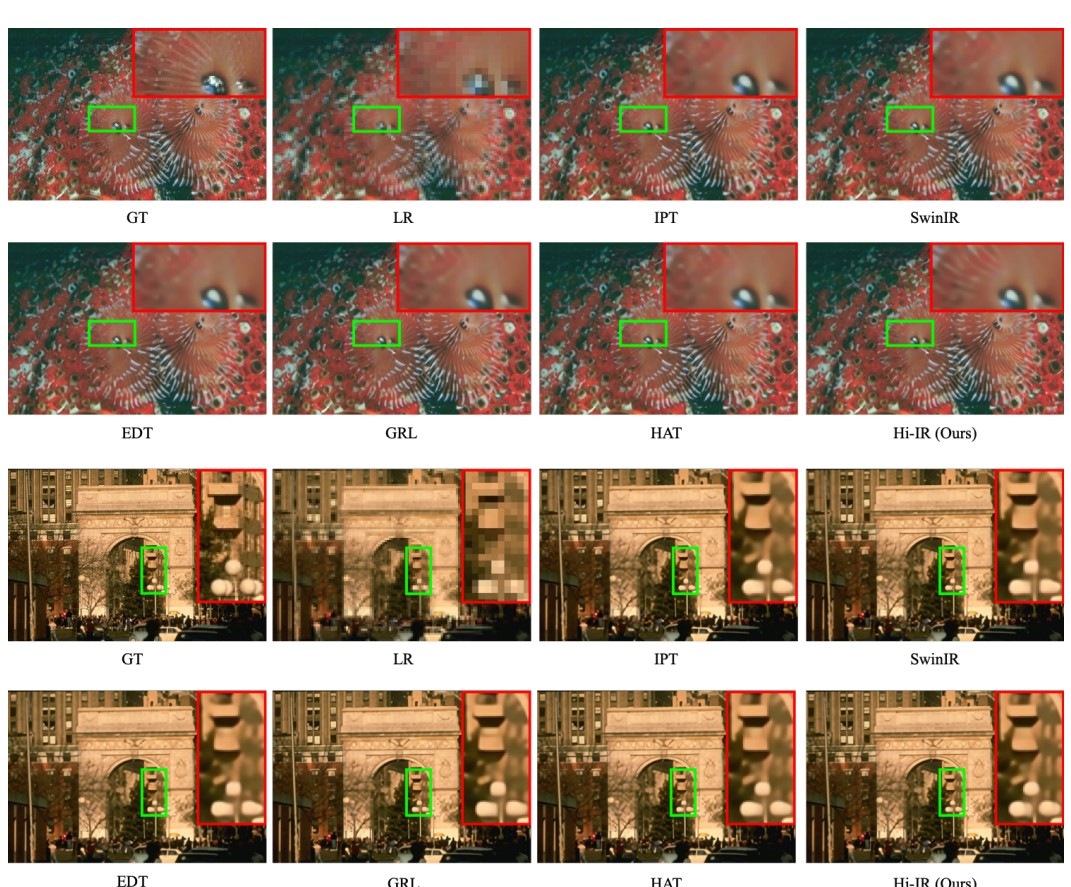

Figure 11: Visual results for classical image SR on B100 dataset. The upscaling factor is ×4.

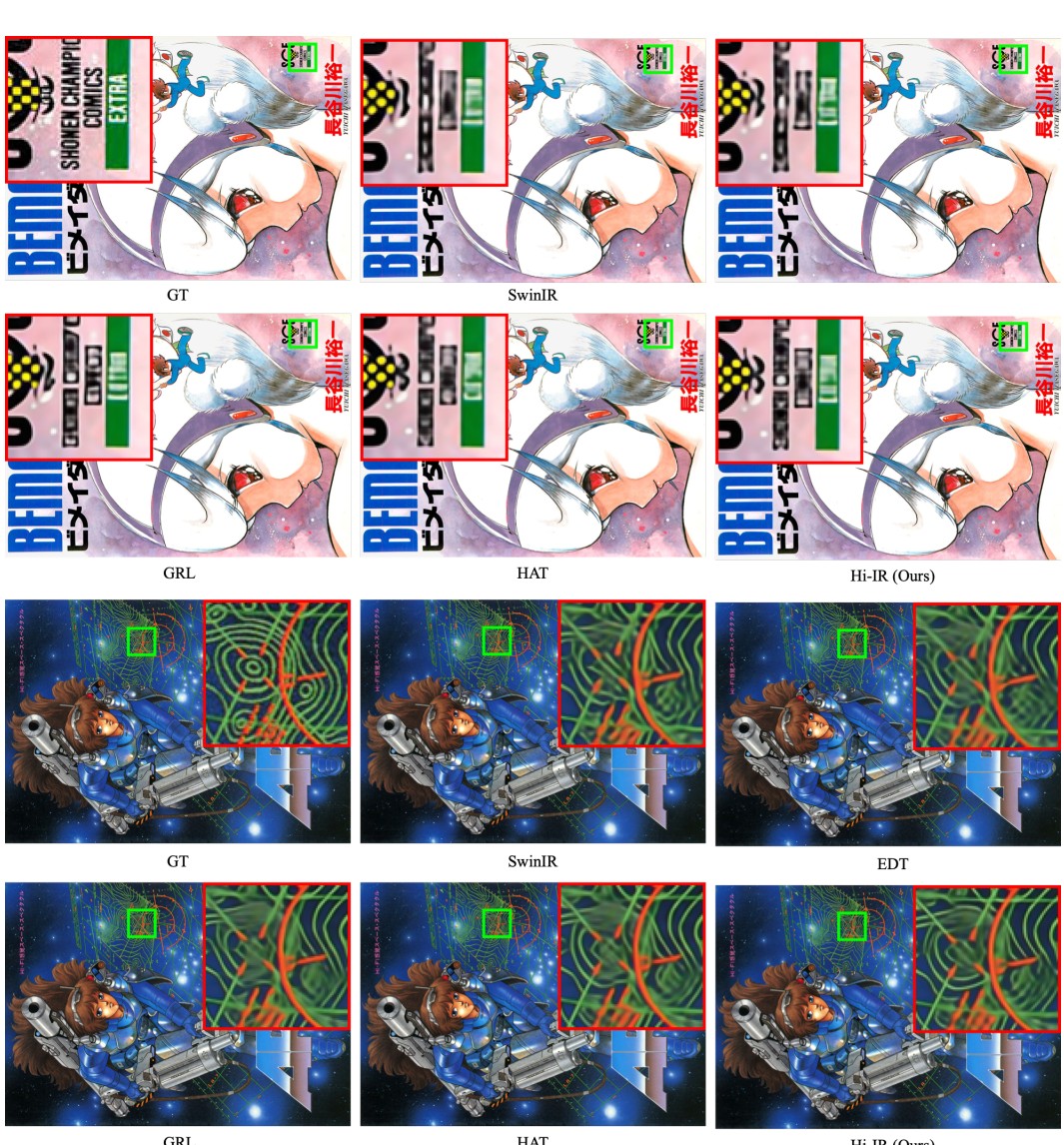

Figure 12: Visual results for classical image ×4 SR on Manga109 dataset.

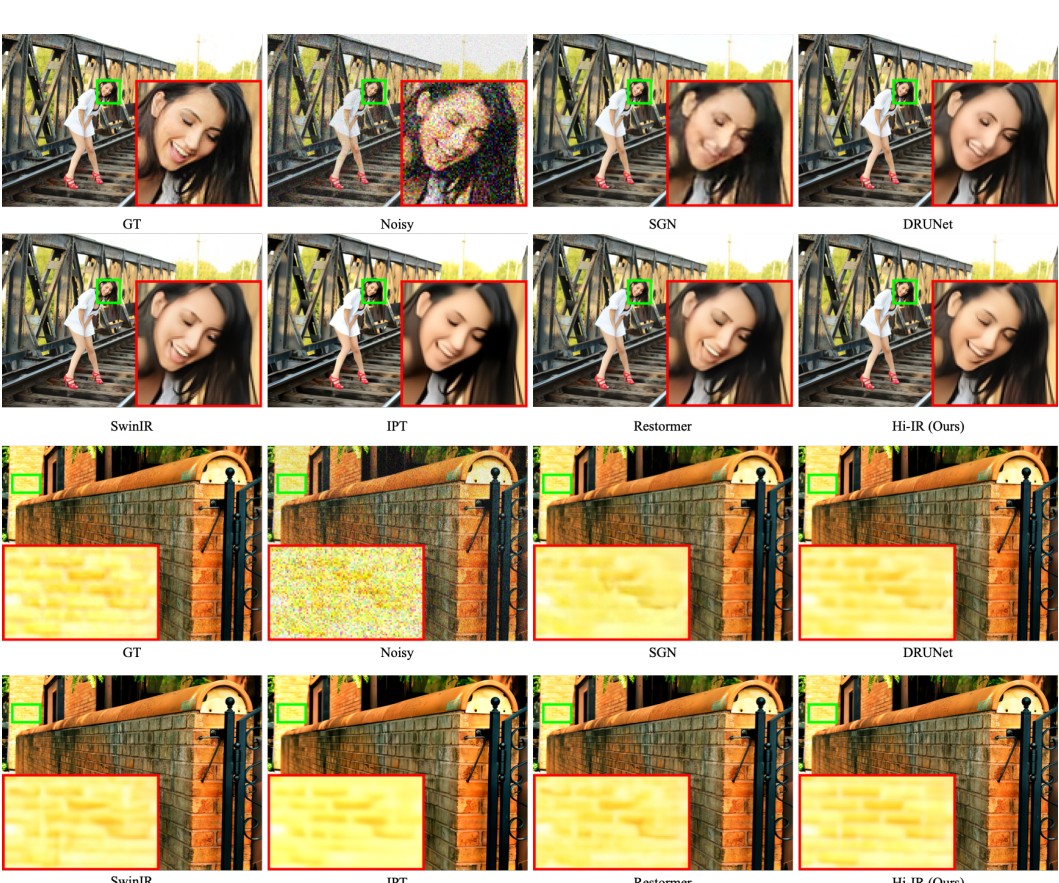

Figure 13: Visual results for classical color image denoising on Urban100 dataset. The noise level is $\sigma = 50$.

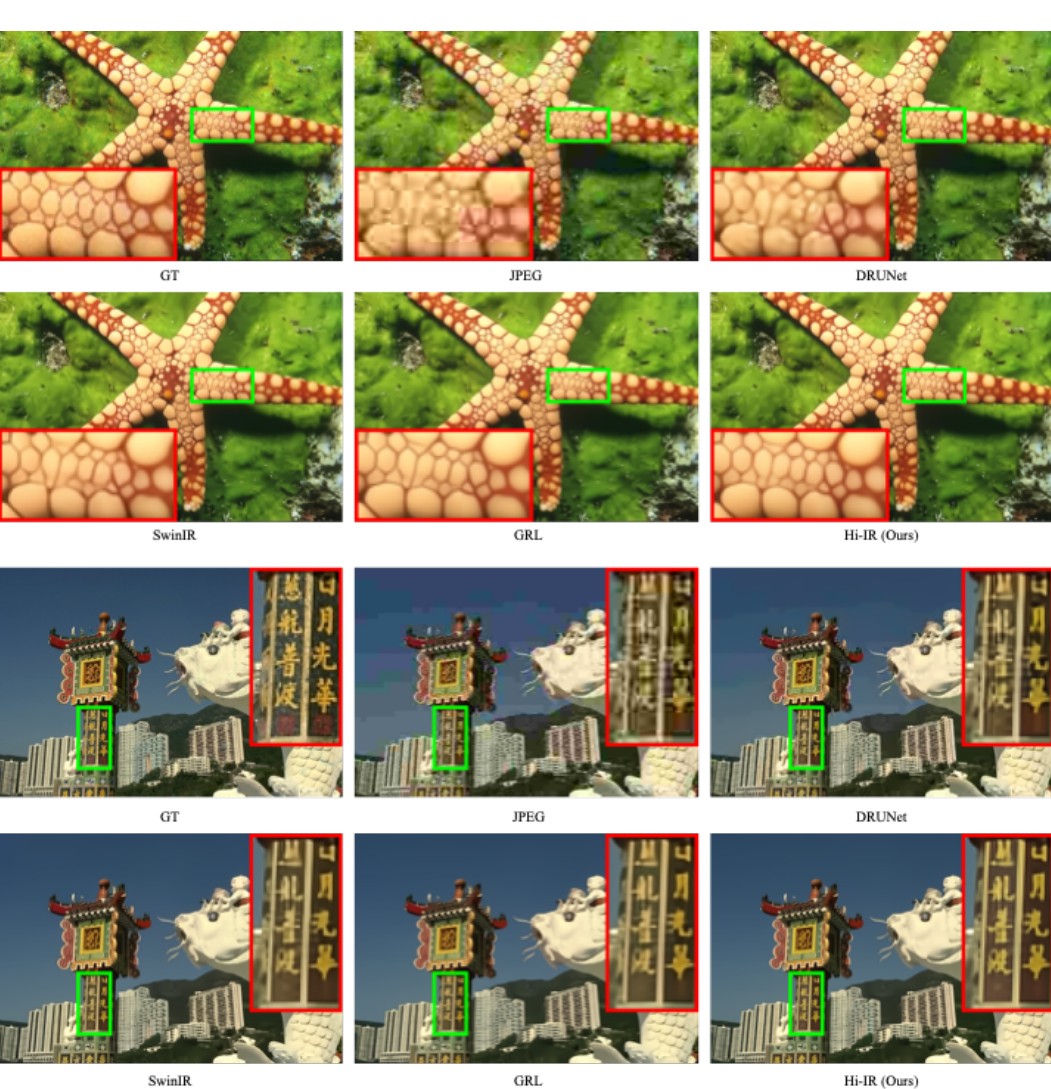

Figure 14: Visual results for color image JPEG compression artifact removal on BSD500 dataset. The quality factor of JPEG image compression is 10.

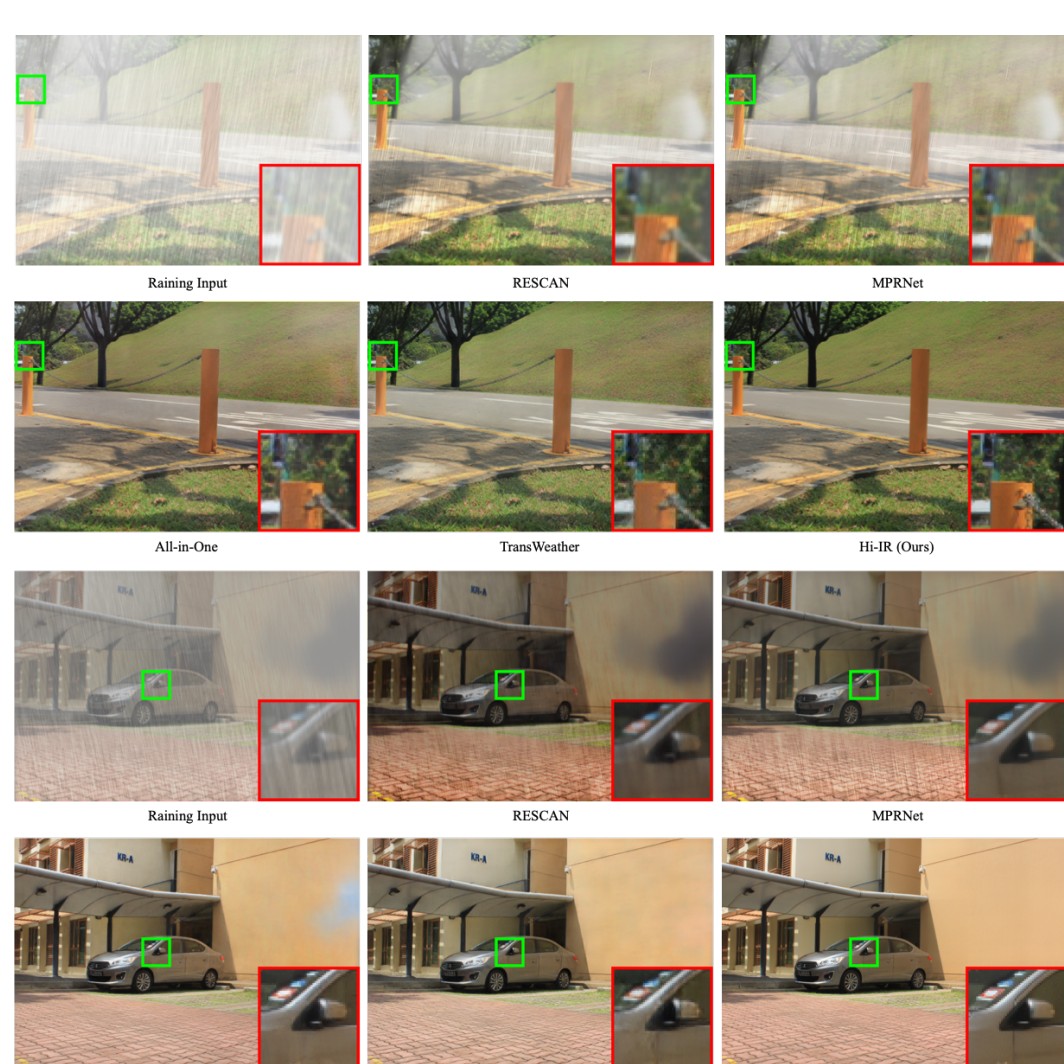

Figure 15: Visual results for restoring images in adverse weather conditions.

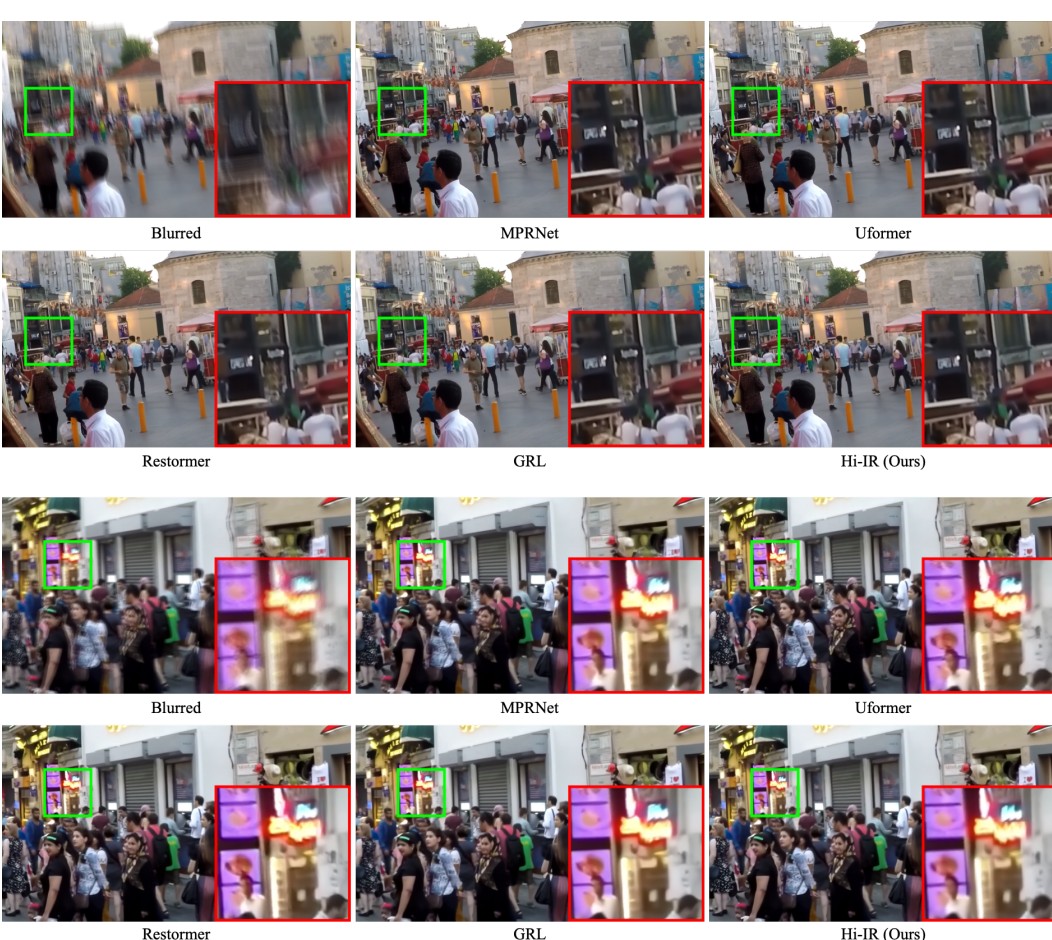

Figure 16: Visual results for single image motion deblurring. The proposed method Hi-IR could recover sharper details compared with the other methods.

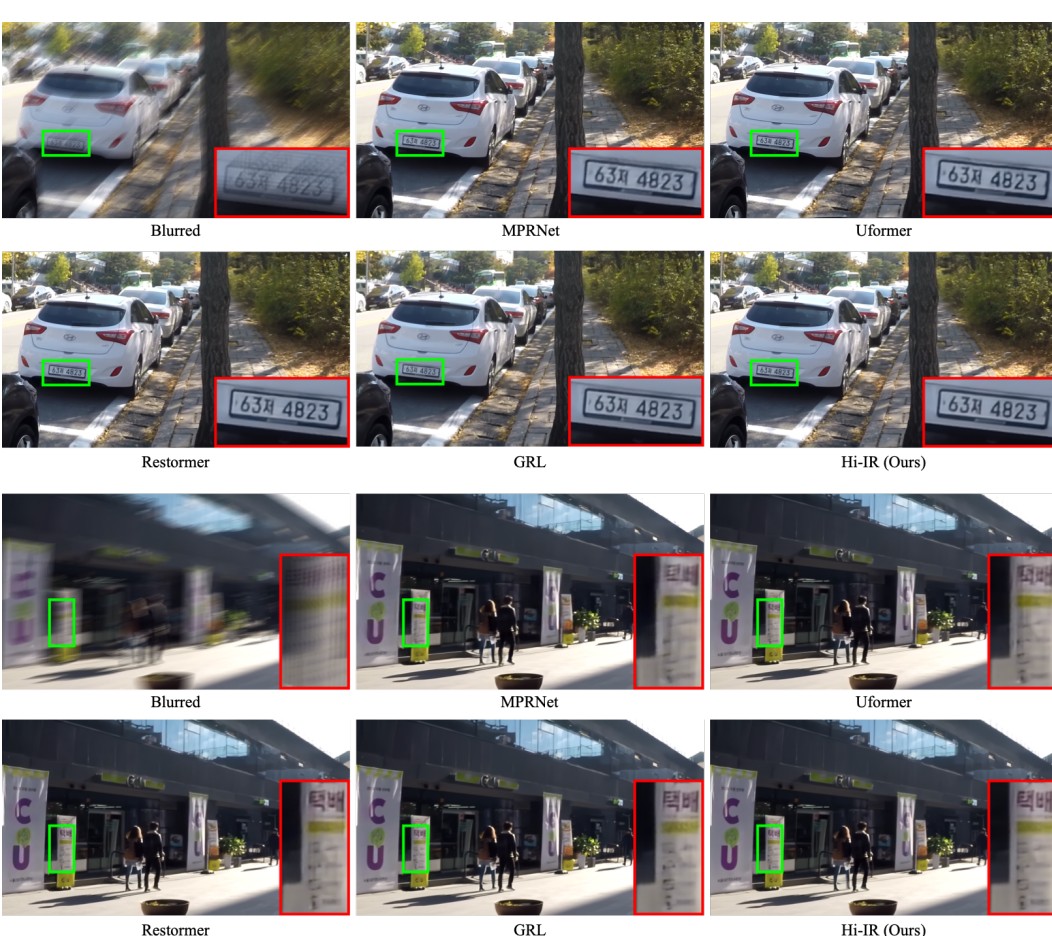

Figure 17: Visual results for single image motion deblurring. The proposed method Hi-IR could recover sharper details compared with the other methods.

