# OpenReview forum: "Hierarchical Information Flow for Generalized Efficient Image Restoration"
_ICLR.cc/2025/Conference — ICLR 2025 Conference Withdrawn Submission_

### Official Review · Reviewer_3HDo · 2024-10-31

**Soundness:** 2
**Presentation:** 2
**Contribution:** 2
**Rating:** 5
**Confidence:** 5

**Summary:**

This paper proposes a hierarchical information tree structure to represent degraded images across three levels, aiming to balance efficiency and model capacity. The hierarchical tree architecture also supports effective model scaling. Extensive experiments are conducted to validate the approach.

**Strengths:**

1. The paper is well-motivated and easy to read.
2. This paper studies the model scaling problem in image restoration, which is valuable.
3. Extensive experiments have been conducted.

**Weaknesses:**

Although the paper presents a compelling narrative, the challenge lies in balancing the scope and complexity of window attention while improving global information propagation efficiency, a topic that has been widely studied in recent years [1]. The proposed solution—window self-attention, permutation, and window self-attention—follows a similar approach as in [2]. These methods are missed and are not discussed. Please explicitly compare the hierarchical information flow mechanism with the random shuffle and spatial shuffle approaches in [1] and [2],and highlight key differences or advantages. Additionally, the paper appears somewhat rushed, with several mistakes. For example, in line 244, "Fig.3(a)" not "Fig.3(c)", and in line 218, "Convplutional" is misspelled.

[1] Random shuffle transformer for image restoration, ICML2023.
[2] Shuffle Transformer: Rethinking Spatial Shuffle for Vision Transformer

**Questions:**

Why are the experiments in Tables 3, 4, 5, 6, and 7 conducted on different datasets?
Why are 2× and 3× scales used in Table 3, while 2× and 4× scales are used in Table 5?
Please provide a brief explanation in the paper for why different datasets and scale factors were chosen for each set of experiments, and how this impacts the interpretation of the results. This would help improve the paper's clarity and consistency.

---

> ### Author Response · Authors · 2024-11-21
> **Author Rebuttal (Part 1 / 2)**
>
> ### Q1: Comparison with ShuffleFormer and Shuffle Transformer.
> ***Ans:*** Thanks a lot for the comments. We compare with Random shuffle transformer [1] and Shuffle transformer [2]. The following comparson is added to Appendix D in the revised paper. Both methods use spatial shuffle operations to facilitate non-local information exchange, with one being random and the other deterministic.
>
> 1. Random Shuffle Transformer (ShuffleFormer) [1] applies random shuffling on the spatial dimension, which increases the probability of global information existing within a local window. While this operation extends the receptive field globally in a single step, it compromises the relevance of pixels within the window. In contrast, the hierarchical information flow proposed in this paper progressively propagates information from local to global while preserving the relevance of attended pixels. A comparison with ShuffleFormer on image deblurring is presented in the following table. Hi-IR outperforms ShuffleFormer by a significant margin while using 55.5% fewer parameters. This demonstrates the effectiveness of the hierarchical information flow method introduced in this work. The comparson is added to Table 11 of the main paper.
>
> | Method        | Model Size | GoPro PSNR | HIDE PSNR |
> |---------------|------------|------------|-----------|
> | ShuffleFormer | 50.61M     | 33.38      | 31.25     |
> | Hi-IR         | 22.33M     | 33.99      | 31.64     |
>
> 2. Shuffle Transformer employs a spatial shuffle operation to aggregate information from distant pixels or tokens. However, it differs from the proposed Hi-IR in several key aspects. First, Shuffle Transformer does not enable progressive information propagation within a hierarchical tree structure. Second, its shuffle operation is based on a fixed grid size of $g = 8$. The distance between pixels in the shuffled window is $H/g$ and $W/g$ along the two axes, which directly depends on the image size. For large images (e.g., 1024 pixels), this design forces distant pixels to attend to one another, often introducing irrelevant information. Consequently, this operation is unsuitable for image restoration tasks, where image sizes can become extremely large. In contrast, the L2 information flow attention proposed in this paper limits the maximum patch size, thereby constraining the maximum distance between pixels at this stage. This restriction enhances the relevance of pixel interactions, making it more effective for image restoration tasks.
>
> [1] Xiao, Jie, et al. "Random shuffle transformer for image restoration." International Conference on Machine Learning. PMLR, 2023.
>
> [2] Huang, Zilong, et al. "Shuffle transformer: Rethinking spatial shuffle for vision transformer." arXiv preprint arXiv:2106.03650 (2021).
>
> ### Q2: Typos
> ***Ans:*** Thanks a lot for pointing out the typos. We have corrected the typos in the revised version.
> ### Q3: Different datasets and scale factors in Tables 3, 4, 5, 6, and 7
> ***Ans:*** Thank you for highlighting the uniformity of the datasets and scaling factors in the ablation study. We provide the following clarifications.
>
> 1. Different test sets in Tables 3, 4, 5, 6, and 7: The ablation study was conducted on super-resolution (SR) tasks, utilizing five test sets: Set5, Set14, BSD100, Urban100, and Manga109. The model was trained once and evaluated on all datasets. However, due to space constraints, we could not display all test results. Notably, the performance gap between different realizations or training methods is more pronounced on Urban100 and Manga109 compared to Set5, Set14, and BSD100. This discrepancy reflects the characteristics of the datasets rather than the methods themselves, as the results are consistent across datasets. To enhance the diversity of reported results, we primarily presented results on the representative dataset Set5 in Tables 4 and 6, Urban100 in Table 7, and across all test sets in Tables 3 and 5.
>
> 2. Different scaling factors in Table 3 and Table 5
> We indeed conducted experiments for all scaling factors in both ablation studies. However, due to space constraints in the main paper, some results were omitted. The amended results are presented below. We also append the full table in Appendix B.

---

> ### Author Response · Authors · 2024-11-21
> **Author Rebuttal (Part 2 / 2): Additional table**
>
> Table 3: Model scaling-up exploration.
> | Scale     | Model size | Warmup | Conv Type | Set5  | Set14 | BSD100 | Urban100 | Manga109 |
> |-----------|------------|--------|-----------|-------|-------|--------|----------|----------|
> | $3\times$ |      15.87 | No     | conv1     | 35.06 | 30.91 |  29.48 |    30.02 |    34.41 |
> | $3\times$ |      57.78 | No     | conv1     |  34.7 | 30.62 |  29.33 |    29.11 |    33.96 |
> | $3\times$ |      57.78 | Yes    | conv1     | 34.91 | 30.77 |  29.39 |    29.53 |    34.12 |
> | $3\times$ |      54.41 | Yes    | linear    | 35.13 | 31.04 |  29.52 |     30.2 |    34.54 |
> | $3\times$ |      55.91 | Yes    | conv3     | 35.14 | 31.03 |  29.51 |    30.22 |    34.76 |
> | $4\times$ |      15.84 | No     | conv1     | 33.00 | 29.11 |  27.94 |    27.67 |    31.41 |
> | $4\times$ |      57.74 | No     | conv1     | 33.08 | 29.19 |  27.97 |    27.83 |    31.56 |
> | $4\times$ |      57.74 | Yes    | conv1     | 32.67 | 28.93 |  27.83 |    27.11 |    30.97 |
> | $4\times$ |      54.37 | Yes    | linear    | 33.06 | 29.16 |  27.99 |    27.93 |    31.66 |
> | $4\times$ |      55.88 | Yes    | conv3     | 33.06 | 29.16 |  27.97 |    27.87 |    31.54 |
>
> Table 5: Dot production attention vs. cosine similarity attention for model scaling. PSNR reported for SR.
> | Scale     | Attn. type  | Set5  | Set14 | BSD100 | Urban100 | Manga109 |
> |-----------|-------------|-------|-------|--------|----------|----------|
> | $3\times$ | cosine sim. | 34.92 | 30.86 |  29.40 |    29.82 |    34.18 |
> | $3\times$ |   dot prod. | 34.98 | 30.98 |  29.45 |    30.06 |    34.35 |
> | $4\times$ | cosine sim. | 33.08 | 29.15 |  27.96 |    27.90 |    31.40 |
> | $4\times$ |   dot prod. | 33.14 | 29.09 |  27.98 |    27.96 |    31.44 |

---

> > ### Author Response · Authors · 2024-11-23
> > **Please let us know if you have additional questions**
> >
> > Dear Reviewer [3HDo](https://openreview.net/forum?id=C0Ubo0XBPn&noteId=lgdQ2JFhQt),
> >
> > Thank you for the comments on our paper.
> >
> > We have provided a response and a revised paper on Openreview based on the comments. Since the discussion phase ends on Nov. 26, we would like to know whether we have addressed all the issues. Please consider raising the scores after this discussion phase.
> >
> > Thank you

---

> > > ### Author Response · Authors · 2024-11-25
> > > **Updated response and PDF file**
> > >
> > > Dear Reviewer [3HDo](https://openreview.net/forum?id=C0Ubo0XBPn&noteId=lgdQ2JFhQt),
> > >
> > > We have updated our response and the PDF file to provide better reference. As the deadline for the discussion phase is approaching, please feel free to let us know if you have any further questions.
> > >
> > > Thank you very much.

---

> > > > ### Author Response · Authors · 2024-11-26
> > > > **Discussion phase**
> > > >
> > > > Dear Reviewer,
> > > >
> > > > As the deadline for the discussion phase will end soon (Nov 26), please let us know whether we have addressed all the questions.
> > > >
> > > > Thank you,

---

> ### Comment · Reviewer_3HDo · 2024-11-26
> **Official Comment by Reviewer 3HDo**
>
> Thank you for your detailed responses.
> The novelty of the proposed information flow is relatively limited, which I believe falls short of the acceptance standards of ICLR. Moreover, the proposed information flow does not demonstrate a clear connection to being "Generalized and Efficient," which I believe is an overstatement.
> I have decided to maintain my rating.

---

### Official Review · Reviewer_VDnP · 2024-11-03

**Soundness:** 2
**Presentation:** 3
**Contribution:** 3
**Rating:** 6
**Confidence:** 4

**Summary:**

This paper proposes an efficient image restoration (IR) method called Hi-IR. Specifically, it introduces a hierarchical information flow that efficiently aggregates global context for each pixel. The authors also scale up the network and offer insights into training larger models. Extensive experiments across seven tasks validate the effectiveness and generalization capability of Hi-IR.

**Strengths:**

1. The experimental results on seven IR tasks validate the strong task generalization ability of the proposed method. The ablation studies also provide evidence of the effectiveness of individual modules.

2. The exploration of scaling IR models may provide practical insights for future research.

3. The writing is clear and easy to follow.

**Weaknesses:**

Major:
1. The details of the permutation operation in Section 3.2 are not fully explained. Various types of permutations could be considered, such as random and circular permutations. I believe this operation is the main technical difference from the window-shift operation used in SwinIR [1]. Additional information on the permutation approach would enhance clarity.

2.From my perspective, the core concept: hierarchical information flow (HIF for short) is essentially a window attention [1] without shifting (L1 information) and a new cross-window interaction operation (L2 information, i.e. permute then MSA). Therefore, the technical novelty is limited.The author may give more discussion on the other realizations of HIF.

3.I believe a complexity analysis of the proposed HIF is beneficial to show the efficiency advantage of Hi-IR. On the other hand, it will provide more insights on how to choose better realizations of HIF (Line 078).

Minor:
1. In the second paragraph of Related Work, the authors introduce several attention mechanisms for IR (Line 131-133), whereas the corresponding reference is missing.

2. The spacing between some captions and corresponding figures/tables (e.g. Figure 5, Table 15) is not well set.

3. Some typos (e.g. Line 016 remove propagation, Line 244 change Fig.3(c) to Fig.3(b)).

[1] Swinir: Image restoration using swin transformer. ICCVW21.

**Questions:**

1. I’m curious about the model's performance when scaled to a larger size (e.g., ~100M parameters). Is the proposed scaling method effective for models of this size?

2. Figure 2 is somewhat unclear. Could you clarify what the different colors represent?

---

> ### Author Response · Authors · 2024-11-21
> **Author Rebuttal**
>
> ### Q1: The details of the permutation operation
> ***Ans:*** We explained the permutation operation in detail in the response to the [common question Q1](https://openreview.net/forum?id=C0Ubo0XBPn&noteId=fc6fA2ngat). The paper is also revised accordingly.
>
> ### Q2: Technical novelty
> ***Ans:*** Our response to this question is detailed in the response to the [common question Q2](https://openreview.net/forum?id=C0Ubo0XBPn&noteId=RC6U0SeUGI). In short, the proposed hierarchical information flow is different from the previous method in the following aspects.
> 1. It propagates information from local to global with controlled computational complexity. Whereas, window attention is mainly conducted for local patches, e.g. $8\times 8$.
> 2. L2 information flow attention propagates information at an elevated level while enhancing pixel relevance by constraining the size of the enlarged region. This capability is not achieved by previous methods like ShuffleFormer and Shuffle Transformer.
> 3. We propose three strategies to systematically guide model scaling-up including removing heavyweight $3\times 3$ convolution, warmup, and using dot product attention. Theoretical analysis of why these strategies work is added to Appendix B.
>
> ### Q3: Complexity analysis of the proposed hierarchical information flow
> ***Ans:*** Thanks a lot for the suggestion. We conducted several analyses on the complexity of the proposed hierarchical information flow in Section 5.1.
> 1. We explored two approaches to implementing L1 and L2 information flow in Transformer layers: alternating them across consecutive layers or integrating both within a single layer. Our findings indicate that integrating L1 and L2 information flow within the same layer enhances performance. The designed Version 3 unifies L1/L2 information flows conceptually into a single flow with an expanded receptive field.
> 2. We examined the impact of tree structure depth, investigating depths of 2, 3, and 4. The results show that increasing the depth improves performance but at the cost of higher computational complexity. Balancing efficiency and accuracy, we selected a tree structure with a depth of 3.
> 3. In addition, we also scale up the model with over 100M parameters. This additional experiment validates the potential to further increase the model size.
>
> ### Q4: Minor points
> ***Ans:*** All the minor points mentioned by the reviewer are addressed in the revised version of this paper.
>
> ### Q5: Further scaling up model to 100M parameters.
> ***Ans:*** Thanks for the comments. The embedding dimension was increased to 256, and the number of Hi-IR transformer stages was set to 12, with 12 transformer layers in each stage. The model was trained for 200k iterations. The experimental results are presented below. While the network has not yet reached full convergence, the early results indicate that training progresses well as the model is further scaled up. Consistent improvements are also observed for the larger model.
>
> | Scale     | Params. | Set5  | Set14 | BSD100 | Urban100 | Manga109 |
> |-----------|---------|-------|-------|--------|----------|----------|
> | $2\times$ |  14.68M | 38.56 | 34.79 |  32.63 |    34.49 |    39.89 |
> | $2\times$ | 110.90M | 38.72 | 35.19 |  32.75 |    35.04 |    40.97 |
>
> ### Q6: Colors in Figure 2
> ***Ans:*** The different colors represent local information. The blending of colors at higher levels in the figure indicates that information gradually propagates beyond the local patch. This information is updated in the caption of this figure.

---

> > ### Author Response · Authors · 2024-11-23
> > **Please let us know if you have additional questions**
> >
> > Dear Reviewer [VDnP](https://openreview.net/forum?id=C0Ubo0XBPn&noteId=C6AMGv84Ux),
> >
> > Thank you for the comments on our paper.
> >
> > We have provided a response and a revised paper on Openreview based on the comments. Since the discussion phase ends on Nov. 26, we would like to know whether we have addressed all the issues. Please consider raising the scores after this discussion phase.
> >
> > Thank you

---

> > > ### Author Response · Authors · 2024-11-25
> > > **Updated response and PDF file**
> > >
> > > Dear Reviewer [VDnP](https://openreview.net/forum?id=C0Ubo0XBPn&noteId=C6AMGv84Ux),
> > >
> > > We have updated our response and the PDF file to provide better reference. As the deadline for the discussion phase is approaching, please feel free to let us know if you have any further questions.
> > >
> > > Thank you very much.

---

> > > ### Comment · Reviewer_VDnP · 2024-11-25
> > >
> > > Thank you for your detailed responses, which have partially addressed my concerns.
> > >
> > > While the authors have provided comparisons of efficiency metrics such as FLOPs, runtime, and parameters. I believe that for a more comprehensive analysis of the complexity associated with hierarchical information flow, it is essential to include a theoretical perspective on both time and space complexity.
> > >
> > > I have decided to maintain my rating.

---

> > > > ### Author Response · Authors · 2024-11-26
> > > > **Author response: space and time complexity**
> > > >
> > > > Thanks a lot for the suggestion. We compared the space and time complexity, and the effective receptive field of the proposed method with a couple of other self-attention methods including global attention and window attention. Suppose the input feature has the dimension $B \times C \times H \times W$, the window size of window attention is $p$, the number of attention heads is $h$, larger patch size of the proposed L2 information flow is $P=s \times p$, the expansion ratio of the MLP in transformer layer is $\gamma$. For the space complexity, we consider the tensors that have to appear in the memory at the same time, which include the input tensor, the query tensor, the key tensor, the value tensor, and the attention map.
> > > >
> > > > The time complexity of the proposed transformer layer is
> > > > $\mathcal{O}\left((5+2\gamma)BHWC^2 + \frac{3}{2}BHWp^2C+\frac{3}{2}BHWs^2C+9\gamma BHWC\right)$. The last term is very small compared with the former two terms, and can be omitted. Thus, the time complexity is simplified as $\mathcal{O}\left((5+2\gamma)BHWC^2 + \frac{3}{2}BHWp^2C+\frac{3}{2}BHWs^2C\right)$.
> > > >
> > > > The space complexity of the proposed transformer layer is
> > > > $\mathcal{O}\left(3BHWC + BHWh\max{(p^2, s^2)}\right)$. The maximum receptive field of two consecutive transformer layers is $16P$.
> > > >
> > > > In the following table, we list the space and time complexity, and receptive field of global attention, window attention, and the proposed method. As shown in this table, window attention is much more efficient than global attenion but with cost of reduced receptive field. The proposed hierarchicial information flow mechanism is more efficient than window attention in propagating information to the global range. As shown in the third row, to achieve the same receptive field as the proposed method, space and time complexity of window attention is much higher than the proposed method.
> > > >
> > > >
> > > >
> > > > | Attn. Method                  | Time Complexity | Space Complexity | Max receptive field of two transformer layers |
> > > > |-------------------------------|-----------------|------------------|-------------------------------------------|
> > > > | Global Attn.                  | $\mathcal{O}\left((4+2\gamma)BHWC^2 + {2}B(HW)^2C\right)$                 | $\mathcal{O}\left(4BHWC + B(HW)^2h\right)$                 |                                        $H \times W$ |
> > > > | Window Attn. ($p \times p$)   | $\mathcal{O}\left((4+2\gamma)BHWC^2 + {2}BHWp^2C\right)$                |  $\mathcal{O}\left(4BHWC + BHWhp^2\right)$                 |                                        $2p \times 2p$ |
> > > > | Window Attn. ($8P \times 8P$) | $\mathcal{O}\left((4+2\gamma)BHWC^2 + {128}BHWp^2s^2C\right)$                | $\mathcal{O}\left(4BHWC + 64BHWhp^2s^2\right)$                 |                                       $16P \times 16P$ |
> > > > | The proposed                  | $\mathcal{O}\left((5+2\gamma)BHWC^2 + \frac{3}{2}BHW(p^2+s^2)C\right)$                | $\mathcal{O}\left(3BHWC + BHWh\max{(p^2, s^2)}\right)$                 |                                      $16P \times 16P$ |

---

### Official Review · Reviewer_wAuz · 2024-11-03

**Soundness:** 3
**Presentation:** 3
**Contribution:** 3
**Rating:** 5
**Confidence:** 5

**Summary:**

This paper proposes a hierarchical information flow principle for general image restoration tasks, which aims to address three significant problems in image restoration including a generalized and efficient IR model, model scaling, and the performance of a single model on different IR tasks. The paper comprehensively analyzes different configurations of model scaling and provides sufficient evaluation results on different image restoration tasks.

**Strengths:**

1. The paper clearly describes the motivations of the proposed method, and the structure of the paper is well-organized, and easy to follow.

2. The paper provides comprehensive experiments to evaluate the performance of different model scaling configurations, which are convincing.

3. The paper demonstrates that the proposed hierarchical information flow design is effective for performance improvement in different IR tasks.

**Weaknesses:**

1. The proposed method appears to have limited novelty. First, it adopts the U-Net structure, whose efficiency has already been evaluated in works like Uformer [1] and Restormer [2]. Second, the hierarchical information flow design seems to be another variation of existing efficient self-attention mechanisms.

2. The claim that generalization and efficiency are inherently a trade-off may not be entirely accurate. Why would a model with strong generalization capabilities be considered inefficient? Is there any external reference or evidence to support this claim?

3. The paper does not clearly explain the details of the tree-based self-attention mechanism. How is the tree structure specifically applied within the self-attention mechanism? Please provide more details or examples to clarify this.

4. The paper claims that the proposed method can handle images with various degradations, but the experiments mainly show its effectiveness on individual IR tasks. Since real-world images often have multiple types of degradations, how does the proposed method perform when applied to such images?

[1] Wang, Zhendong, Xiaodong Cun, Jianmin Bao, Wengang Zhou, Jianzhuang Liu, and Houqiang Li. "Uformer: A general u-shaped transformer for image restoration." In Proceedings of the IEEE/CVF conference on computer vision and pattern recognition, pp. 17683-17693. 2022.

[2] Zamir, Syed Waqas, Aditya Arora, Salman Khan, Munawar Hayat, Fahad Shahbaz Khan, and Ming-Hsuan Yang. "Restormer: Efficient transformer for high-resolution image restoration." In Proceedings of the IEEE/CVF conference on computer vision and pattern recognition, pp. 5728-5739. 2022.

**Questions:**

1. It is better to clarify what distinguishes your approach from these prior works.

2. Why would a model with strong generalization capabilities be considered inefficient? Is there any external reference or evidence to support this claim?

3.  How is the tree structure specifically applied within the self-attention mechanism? Please provide more details or examples to clarify this.

4. How does the proposed method perform when applied to real-world images for image super-resolution and image denoising? More experiment results should be included.

---

> ### Author Response · Authors · 2024-11-21
> **Author Rebuttal**
>
> ### Q1: Novelty of the paper
> ***Ans:*** Thanks a lot for the comment. We address the reviewer's concern in the response to the [common question Q2](https://openreview.net/forum?id=C0Ubo0XBPn&noteId=RC6U0SeUGI). We highlight several points in the following. First, although we used columnar architecture for SR and UNet architecture for the other tasks, we did not claim that the U-Net architecture is a contribution of this paper. Those are standard choices following the literature. Second, the hierarchical information flow proposed in this paper can be regarded as an efficient self-attention mechanism. The new mechanism has the following advantages: 1) propagating information from local to global in a progressive and efficient way; 2) by constraining the practical patch size in each level, the proposed method promotes the relevance of pixels in the same level, which can be a problem for ShuffleFormer and Shuffle Transformer. The detailed comparison with ShufflerFormer and Shuffle Transformer is done as suggested by [Reviewer 3HDo](https://openreview.net/forum?id=C0Ubo0XBPn&noteId=lgdQ2JFhQt). Third, the main contribution of Uformer is validating the UNet architecture for Transformers while the main contribution of Restormer is the self-attention along the channel dimension. Both of them are different from the proposed hierarchical information flow in this paper.
>
> ### Q2: Claim of generalization and efficiency
> ***Ans:*** Thank you for the valuable feedback. Our original intention was to convey that generalizing to different IR tasks requires careful consideration of the unique properties of each task. Simply combining computational mechanisms designed for different IR tasks does not necessarily result in an efficient solution. We have revised the statements in the paper to better reflect this perspective.
>
> ### Q3: Details of the tree-based information flow
> ***Ans:*** Thanks for the comment. We explain the details of the tree-based information flow and the corresponding permutation operation in the response to the [common question Q1](https://openreview.net/forum?id=C0Ubo0XBPn&noteId=fc6fA2ngat). We also updated the paper accordingly.
>
> ### Q4: Handling images with various degradations
> ***Ans:*** We thank the reviewer for the suggestions. To validate the generalization capability of the proposed method to different types of degradation, we conducted the following experiments. First, we used the same model for both denoising and JPEG compression artifact removal tasks. Notably, a single model was trained to handle varying levels of degradation. Second, we performed experiments on image restoration under adverse weather conditions, including rain, fog, and snow. Third, we further investigated a one-in-all image restoration setup, encompassing five different tasks with real-world images. The experimental results demonstrate that the proposed method outperforms previous methods by a significant margin. These three sets of experiments collectively highlight that the proposed hierarchical information flow mechanism enables training a single model that generalizes effectively to various types and levels of degradation. We updated the experiments and results in Appendix C.4.
>
> | Method       | Params. | Dehazing | Deraining | Denoising | Deblurring | Low-Light | Average |
> |--------------|---------|----------|-----------|-----------|------------|-----------|---------|
> | AirNet       | 9M      | 21.04    | 32.98     | 30.91     | 24.35      | 18.18     | 25.49   |
> | IDR          | 15M     | 25.24    | 35.63     | 31.6      | 27.87      | 21.34     | 28.34   |
> | PromptIR     | 33M     | 26.54    | 36.37     | 31.47     | 28.71      | 22.68     | 29.15   |
> | AdaIR        | 29M     | 30.53    | 38.02     | 31.35     | 28.12      | 23.00        | 30.20    |
> | Hi-IR (Ours) | 22M     | 31.42    | 38.67     | 31.58     | 28.95      | 23.12     | 30.75   |

---

> > ### Author Response · Authors · 2024-11-23
> > **Please let us know if you have additional questions**
> >
> > Dear Reviewer [wAuz](https://openreview.net/forum?id=C0Ubo0XBPn&noteId=2xbtXsJdqy),
> >
> > Thank you for the comments on our paper.
> >
> > We have provided a response and a revised paper on Openreview based on the comments. Since the discussion phase ends on Nov. 26, we would like to know whether we have addressed all the issues. Please consider raising the scores after this discussion phase.
> >
> > Thank you

---

> > > ### Author Response · Authors · 2024-11-25
> > > **Updated response and PDF file**
> > >
> > > Dear Reviewer [wAuz](https://openreview.net/forum?id=C0Ubo0XBPn&noteId=2xbtXsJdqy),
> > >
> > > We have updated our response and the PDF file to provide better reference. As the deadline for the discussion phase is approaching, please feel free to let us know if you have any further questions.
> > >
> > > Thank you very much.

---

> > > > ### Author Response · Authors · 2024-11-26
> > > > **Discussion phase**
> > > >
> > > > Dear Reviewer,
> > > >
> > > > As the deadline for the discussion phase will end soon (Nov 26), please let us know whether we have addressed all the questions.
> > > >
> > > > Thank you,

---

> > > > > ### Comment · Reviewer_wAuz · 2024-11-26
> > > > >
> > > > > Thank you for the authors' responses. After carefully reading the responses and the revised version, I believe that the proposed hierarchical information flow is a variant of the self-attention mechanism which performs non-local operations to aggregate similar patches. It should be discussed and compared with the difference between the proposed method and other self-attention mechanisms. In addition,  the proposed method achieves slight performance improvement on the benchmark datasets of several tasks, such as single image super-resolution, and denoising.  Therefore, I would like to maintain my original score and rating.

---

### Official Review · Reviewer_99w8 · 2024-11-04

**Soundness:** 3
**Presentation:** 2
**Contribution:** 2
**Rating:** 5
**Confidence:** 4

**Summary:**

This paper introduces Hi-IR, a hierarchical information flow mechanism structured as a tree for image restoration. In Hi-IR, information moves progressively from local areas, aggregates at multiple intermediate levels, and then spreads across the entire sequence. By using this hierarchical tree design, Hi-IR eliminates long-range self-attention, enhancing computational efficiency and memory usage. Extensive experiments across seven image restoration tasks demonstrate the effectiveness and generalizability of Hi-IR.

**Strengths:**

- The motivation for this paper is twofold: (1) information flow plays a pivotal role in decoding low-level features, and (2) it is not always necessary to implement information flow through fully connected graphs. Proposing hierarchical information flow via tree-structured attention is a reasonable approach to balancing complexity and the efficiency of global information propagation.
- The paper proposes a model scaling-up solution, such as replacing heavyweight $3 \times 3$ convolutions with lightweight operations. The authors reasonably justify why this approach is effective in Section 4 and Appendix B.1.

**Weaknesses:**

- The authors state that the simple permutation operation facilitates the distribution of $l_1$ information nodes across all windows (L238-239); however, there is no specific explanation of how the permutation is applied. Could the authors answer to the questions below?
   1. Specify exactly which components the permutation is applied to
   2. Clarify if the permutation is random or deterministic
   3. Provide an ablation study isolating the impact of the permutation compared to the tree structure

   This would provide valuable insight into how the permutation contributes to the model's performance.
- The authors mention that the actual implementation of the tree structure, such as the depth of the tree and the number of child nodes, can be configured to ensure computational efficiency (L199-201). However, no accompanying ablation study has been conducted on these parameters. The authors are required to do below:
   1. Conduct experiments varying the tree depth and number of child nodes
   2. Show how these choices impact both performance and computational efficiency
   3. Discuss how different configurations balance local and global information modeling

   This would provide concrete evidence for the effectiveness of the hierarchical design.
- Additional feedbacks:
   - It is essential to provide specific details for the clarity. For example:
     1. Add SR task specification to Table 6 caption
     2. Clarify in main text whether Hi-IR-B or Hi-IR-L is being referred to (Section 5)
     3. Add reference to Table 7 when discussing efficiency analysis (L409-L412)
     4. Spell out "Dn" as "denoising (Dn)" on first use (Table 7)
   - There is a typographical error in the sentence (L257) where the first letter should be capitalized. "for each" should be changed to "For each.”
   - There is a sentence fragment that lacks a main clause (L375-L376).
   - Can authors show the GT in Figure 5?
   - The proposed Hi-IR does not outperform all methods in every instance; therefore, the expression "the proposed Hi-IR outperforms all other comparison methods under both settings" (L474-L475) should be softened.
   - There appear to be discrepancies between the experimental results in the table and the results described in the text in Section 5.2. Could the authors verify this?

**Questions:**

- The reviewer has two questions related to model scaling-up. First, could the authors specify which part of the proposed Hi-IR structure involves replacing heavyweight $3 \times 3$ convolutions with lightweight operations? Second, could the authors explain why applying warming-up and using dot product instead of cosine similarity attention leads to improved scaling-up? Simply showing improved performance with a larger model is insufficient to demonstrate they contribute effective scaling-up convincingly.
- In the paragraph discussing the effect of L1 and L2 information flow (L374-L376), could the authors explain why v3 is superior among v1, v2, and v3? Alternatively, could the authors explain the rationale behind the experimental design for v1 to v3? Simply showcasing the best one among the three designs does not provide informative insight for the reader.

---

> ### Author Response · Authors · 2024-11-21
> **Author Rebuttal (Part 1 / 1)**
>
> ### Q1: Permutation operation
> ***Ans:*** Thanks for the suggestion. The detail of the permutation operation is explained in the response to the [common question Q1](https://openreview.net/forum?id=C0Ubo0XBPn&noteId=fc6fA2ngat). The permutation is done in a deterministic way. Removing the permuation causes the L2 information flow to degrade into L1 information flow, which isolates information as discussed in Line 52 and Table 1 of the main paper. As requested, we did an ablation study to show this effect. The following abation study shows that the permuation operation is important to maintain the performance of the network.
>
> | Dataset          | Set5  | Set14 | BSD100 | Urban100 | Manga109 |
> |------------------|-------|-------|--------|----------|----------|
> | With Permutation | 38.56 | 34.79 | 32.63  | 34.49    | 39.89    |
> | W/O Permutation  | 38.51 | 34.73 |  32.59 |    34.21 |    39.63 |
>
> ### Q2: Depth of tree structure and model complexity
>
> ***Ans:*** Thanks for the insighful comments. Actually, we have done this ablation study to investigate the complexity and efficiency of the proposed tree structure in Section 5.1. And the result is shown in Table 6. To summarize, we have done the following experiments.
> 1. We investigated three versions of the information flow mechanism with different complexities in Figure 4. In Version 1, we use L1 information flow and L2 information flow alternatively in the consecutive Transformer layers. In Version 2 and 3, both L2 and L3 information flow are implemented in a single Transformer Layer. Version 3 differs from Version 2 in that the projection layer between the L1 and L2 attention is removed and the embedding dimension of $Q$ and $K$ is reduced by half. This leads to models with different complexities including Version 1 (\~14.5M), Version 2 (\~19.37M), and Version 3 (\~15.84M).
> Compared with Version 1, two consecutive Transformer Layers in Version 2 and Version 3 have two more information flow attention operations, which can be regarded as a deeper tree. Yet, the performance of Version 2 becomes worse with increased model complexity, which is due to the design. By contrast, Version 3 works better than Version 1 with deepened tree structure and increased model complexity. Considering the tradeoff between PSNR gains and model complexity, we use Version 3 throughout this paper. *In short, well-designed deeper tree structures lead to improved model performance but with increased model complexity.*
> 2. In addition to the investigation of L1 and L2 information flow, we also ablate the effects of L3 information flow. Thus, we remove L3 information flow from the network for all the three versions mentioned above, which leads to tree structures with reduced depth. In particular, the depth of the tree is reduces to 2 for the Version 1 model. By comparing the *'with L3'* and *'w/o L3'* columns of Table 3, we conclude the importance of L3 information flow.
> 3. Moreover, during the rebuttal phase, we implemented a deeper tree structure similar to Version 1. We added another information flow attention. Thus, three information flow attention operations alternate in the network, and propagate information in a $8\times 8$, $64 \times 64$, and $256 \times 256$ patch, respectively.  The experimental results for the tree structure with depth 4 and another two tree structures with depth 2 and 3 discussed above are shown. The performance of the tree structure with depth of 4 is improved but the model size is also increased.
>
> | Tree depth          | 2     | 3     | 4     |
> |----------------|-------|-------|-------|
> | $2\times$             | 38.31 | 38.34 | 38.41 |
> | Model size [M] | 11.87 | 14.35 | 17.19 |
> |                |       |       |       |
> | $4\times$             | 32.85 | 32.89 | 32.95 |
> | Model size [M] | 12.02 | 14.50 | 17.34 |
>
> ### Q3: Addtional feedback
> ***Ans:***: All the addtional feebacks are addressed in the revised version.

---

> ### Author Response · Authors · 2024-11-21
> **Author Rebuttal (Part 2 / 2)**
>
> ### Q4: Model scaling-up strategies
> ***Ans:***: Thanks for the comments. We provide deeper analysis as follows. We also updated the analysis in Section B.2 and B.3 of the Appendix.
> 1. The heavyweight $3 \times 3$ convolutions appears at the end the the Hi-IR stage as shown in Fig. 3c.
> 2. *Warmup* is effective for training large models primarily because it mitigates issues related to unstable gradients and helps the optimizer gradually adapt to the model's large parameter space [1,2]. In the early stages of training, the model's parameters are initialized randomly. A high learning rate at this stage can cause large updates, leading to unstable or divergent training due to exploding or vanishing gradients. Warmup starts with a small learning rate and gradually increases it, allowing the optimizer to find a stable path in the loss landscape before applying larger updates. Warmup enables the model to adapt gradually, avoiding overshooting minima and ensuring smoother convergence.
> 3. *Dot product v.s. cosine similarity*: We analyze the gradient of dot product and cosine similary as follows. Suppose $\mathbf{q}$ denotes the query and $\mathbf{k}$ denotes the keys. Then dot product and cosine similarity between $\mathbf{q}$ and $\mathbf{k}$ are denoted as $\text{dot\\_prod}(\mathbf{q}, \mathbf{k})$, $\text{cos\\_sim}(\mathbf{q}, \mathbf{k})$.
>
>     - The gradient of dot product with respect to $\mathbf{q}$ is denoted as  $\frac{\partial}{\partial \mathbf{q}} \text{dot\\_prod}(\mathbf{q}, \mathbf{k}) = \mathbf{k}$.
>     - The gradient of cosine similarity with respect to $\mathbf{q}$ is $\frac{\partial}{\partial \mathbf{q}} \text{cos\\_sim}(\mathbf{q}, \mathbf{k})
>          = \frac{\mathbf{k}}{\|\mathbf{q}\| \|\mathbf{k}\|} - \frac{(\mathbf{q} \cdot \mathbf{k}) \mathbf{q}}{\|\mathbf{q}\|^3 \|\mathbf{k}\|}
>          = \frac{1}{\|\mathbf{q}\|} \left(\mathbf{\hat{k}} - \text{cos\\_sim}(\mathbf{q}, \mathbf{k}) \mathbf{\hat{q}}\right)$, where $\mathbf{\hat{q}}$ and $\mathbf{\hat{k}}$ are normalized $\mathbf{q}$ and $\mathbf{k}$.
>     The gradients with respect to $\mathbf{k}$ have the similar form. The gradient of cosine similarity involves more terms compared to the gradient of the dot product. This increased complexity in the gradient of cosine similarity makes it more prone to producing large or even unstable gradient values. We conducted a numerical analysis of the gradient values for the two attention methods, with the results presented in Figure 9 of the Appendix. As shown in the figure, the gradient of cosine similarity is indeed more prone to producing large values. This issue becomes more pronounced as the model scales up.
>
> [1] Goyal, P. "Accurate, large minibatch SG D: training imagenet in 1 hour." arXiv preprint arXiv:1706.02677 (2017).
>
> [2] Kalra, Dayal Singh, and Maissam Barkeshli. "Why Warmup the Learning Rate? Underlying Mechanisms and Improvements." arXiv preprint arXiv:2406.09405 (2024).
>
> ### Q5: Rationale for model design in ablation study
> ***Ans:***: Thanks a lot for the question. We align the text better with the aim. This ablation serves two purposes: 1) exploring a better design to place the L1/L2/L3 information flow mechanisms; 2) investigating the influence of the tree depth. When designing the L1/L2 information flow attention mechanism, we need to decide whether to interleave L1/L2 information flow across Transformer layers or to implement them in the same layer. To validate this choice, we developed Version 1 and Version 2. However, Version 2 demonstrated reduced performance despite increased model complexity. To address this issue, we introduced Version 3, where L1 and L2 information flows can be conceptually unified into a single flow with a larger receptive field. We revised the corresponding texts in Section 5.1.

---

> ### Author Response · Authors · 2024-11-23
> **Please let us know if you have additional questions**
>
> Dear Reviewer [99w8](https://openreview.net/forum?id=C0Ubo0XBPn&noteId=jnwAz6NPkd),
>
> Thank you for the comments on our paper.
>
> We have provided a response and a revised paper on Openreview based on the comments. Since the discussion phase ends on Nov. 26, we would like to know whether we have addressed all the issues. Please consider raising the scores after this discussion phase.
>
> Thank you

---

> > ### Author Response · Authors · 2024-11-25
> > **Updated response and pdf file**
> >
> > Dear Reviewer [99w8](https://openreview.net/forum?id=C0Ubo0XBPn&noteId=jnwAz6NPkd),
> >
> > We have updated our response and the PDF file to provide better reference. As the deadline for the discussion phase is approaching, please feel free to let us know if you have any further questions.
> >
> > Thank you very much.

---

> > > ### Author Response · Authors · 2024-11-26
> > > **Discussion phase**
> > >
> > > Dear Reviewer,
> > >
> > > As the deadline for the discussion phase will end soon (Nov 26), please let us know whether we have addressed all the questions.
> > >
> > > Thank you,

---

### Author Response · Authors · 2024-11-21
**Author Response to Common Questions (Part 1 / 2)**

We sincerely appreciate the reviewers' efforts in evaluating our work and their positive feedback on various aspects of our work (**clear motivation, balancing complexity and efficiency, model scaling-up solution, comprehensive experiments, task generalization ability**, *etc*.) We thank the reviewers for their valuable comments and insightful suggestions, which helped us a lot to improve the quality of our paper. Below, we address the common questions raised by the reviewers.

### Q1: Details of tree-based information flow attention and permutation. (Reviewer [99w8](https://openreview.net/forum?id=C0Ubo0XBPn&noteId=jnwAz6NPkd) Q1, Reviewer [wAuz](https://openreview.net/forum?id=C0Ubo0XBPn&noteId=2xbtXsJdqy) Q3, Reviewer [VDnP](https://openreview.net/forum?id=C0Ubo0XBPn&noteId=C6AMGv84Ux) Q1)

***Ans:*** We provide more details about the the proposed mechanism and permutation operations here. Suppose the input tensor of the L2 information flow is $Y^{l_1}\in \mathbb{R}^{H \times W \times C}$. The permutation operation helps to form the hierachical tree described in the paper. The purpose of L2 information flow is to expand the receptive field beyond a local patch with maintained computational efficiency. Two coupled operations are done to serve this purpose.

**First**, as indicated conceptually in Fig. 2(d), $s \times s$ non-overlapping local patches $p \times p$ in L1 information flow are grouped together to form a larger patch with dimension $P \times P$, where $P = sp$. We do not expand to the whole image in this phase due to two considerations: 1) The computational complexity of attention in the global image can be quite high; 2) Not all global image information is relevant to the reconstruction of a specific pixel.

**Second**, after the grouping operation, self-attention is applied to pixels located within the larger patch $P \times P$ but distributed across the small patches $p \times p$. To faciliate the self-attention, the $s\times s$ dispersed pixels need to be grouped together via a permutation operation.

**In short**, the seemingly complicated operation can be done easily by a reshape and a permutation operation. The input tensor is first reshaped to $\hat{Y}^{l_1}\in \mathbb{R}^{\frac{H}{P} \times s \times p \times \frac{W}{P} \times s \times p \times C}$. Then a permuation is done to form $(Y')^{l_1}\in \mathbb{R}^{(\frac{H}{P} \times \frac{W}{P} \times p^2) \times s^2 \times C}$. L1 information flow attention is done within the $p\times p$ patch while L2 information flow attention is conducted among the dispersed pixel locations in the partitioned larger $P \times P$ regions. We also made this clearer in the revised version of the paper.

---

> ### Author Response · Authors · 2024-11-21
> **Author Response to Common Questions (Part 2 / 2)**
>
> ### Q2: Novelty of the paper. (Reviewer [wAuz](https://openreview.net/forum?id=C0Ubo0XBPn&noteId=2xbtXsJdqy) Q1, Reviewer [VDnP](https://openreview.net/forum?id=C0Ubo0XBPn&noteId=C6AMGv84Ux) Q2)
>
> ***Ans:*** The novelty of the paper comes with the follwing aspects.
> 1. We propose a hierachical information flow mechanism to efficiently and progressively propagate information from local to the global field. This is different from previous works such as SwinIR[1], Uformer[2], Restormer[3], ShuffleFormer[4], Shuffle Transformer[5], etc.
>     - The proposed L2 information flow attention facilitates the propagation of information between dispersed pixels within an enlarged region and at an elevated level. Specifically, we avoid expanding to the entire image at this stage for two reasons: 1) the computational complexity of attention across the global image is prohibitively high; and 2) not all global image information is relevant to the reconstruction of a specific pixel. Compared with SwinIR[1], the receptive field is larger. Compared with ShuffleFormer[4] and Shuffle Transformer[5], this mechanism promotes pixel relevance.
>     - The main contribution of Uformer[2] is validating the UNet architecture for Transformers while the main contribution of Restormer[3] is the self-attention along the channel dimension. Both of them are different from the proposed hierarchical information flow in this paper.
>     - Although we use Columnar architecture for SR and UNet architecture for the other tasks, we do not claim the general architecture as the contribution of this paper. Those are standard choices following the literature.
> 2. We conduct thorough experiments and analysis to study model scaling-up for image restoration. We propose three strategies for IR mdoel scaling-up including removing heavyweight $3\times 3$ convolution, warmup, and using dot production for self-attention. Both experimental results and theoretical analysis (Appendix B) is done. In particular:
>     - Removing heavyweight $3\times 3$ convolution from the network avoids intialized weight parameters with small values which leads to vanishing gradients and slow convergence.
>     - Warmup helps because it mitigates issues related to unstable gradients in the early phase of training and helps the optimizer gradually adapt to the model’s large parameter space.
>     - Dot product attention works better than cosine similarity attention becase the gradient of cosine similarity is more prone to producing large or even unstable values.
> 3. We conducted experiments on various image restoration problems including image super-resolution, denoising, motion deblurring, defocus debluring, removing rain, fog, haze, and snow from the image. The thorough analysis validated the generalizability of the proposed method.
>
> [1] Liang, Jingyun, et al. "Swinir: Image restoration using swin transformer." Proceedings of the IEEE/CVF international conference on computer vision. 2021.
>
> [2] Wang, Zhendong, et al. "Uformer: A general u-shaped transformer for image restoration." Proceedings of the IEEE/CVF conference on computer vision and pattern recognition. 2022.
>
> [3] Zamir, Syed Waqas, et al. "Restormer: Efficient transformer for high-resolution image restoration." Proceedings of the IEEE/CVF conference on computer vision and pattern recognition. 2022.
>
> [4] Xiao, Jie, et al. "Random shuffle transformer for image restoration." International Conference on Machine Learning. PMLR, 2023.
>
> [5] Huang, Zilong, et al. "Shuffle transformer: Rethinking spatial shuffle for vision transformer." arXiv preprint arXiv:2106.03650 (2021).

---

### Note · Authors · 2025-02-19

I have read and agree with the venue's withdrawal policy on behalf of myself and my co-authors.

---

### Meta-Review · Area_Chair_M8S2 · 2024-12-17

**Metareview:**

The paper proposes a hierarchical information flow mechanism for image restoration. However, all the reviewers believe that the novelty of the proposed information flow is relatively limited, which falls short of the acceptance standards of ICLR. Besides, the paper lacks a detailed discussion and comparison with other self-attention mechanisms.
Based on the reviewer's average rating, I believe that this paper is not yet ready for publication at ICLR.

**Additional Comments On Reviewer Discussion:**

All the reviewers participated in the discussion, but none of them changed their scores. The average score of the paper remains below the acceptance threshold.

---

### Decision · Program_Chairs · 2025-01-22

Reject